# Even your Teacher Needs Guidance: Ground-Truth Targets Dampen Regularization Imposed by Self-Distillation

**Kenneth Borup**
Department of Mathematics
Aarhus University
kennethborup@math.au.dk

**Lars N. Andersen**
Department of Mathematics
Aarhus University
larsa@math.au.dk

## Abstract

Knowledge distillation is classically a procedure where a neural network is trained on the output of another network along with the original targets in order to transfer knowledge between the architectures. The special case of self-distillation, where the network architectures are identical, has been observed to improve generalization accuracy. In this paper, we consider an iterative variant of self-distillation in a kernel regression setting, in which successive steps incorporate both model outputs and the ground-truth targets. This allows us to provide the first theoretical results on the importance of using the weighted ground-truth targets in self-distillation. Our focus is on fitting nonlinear functions to training data with a weighted mean square error objective function suitable for distillation, subject to $\ell_2$ regularization of the model parameters. We show that any such function obtained with self-distillation can be calculated directly as a function of the initial fit, and that infinite distillation steps yields the same optimization problem as the original with amplified regularization. Furthermore, we provide a closed form solution for the optimal choice of weighting parameter at each step, and show how to efficiently estimate this weighting parameter for deep learning and significantly reduce the computational requirements compared to a grid search.

## 1 Introduction

Knowledge distillation, most commonly known from Hinton et al. (2015), is a procedure to transfer *knowledge* from one neural network (teacher) to another neural network (student).[1] Often the student has fewer parameters than the teacher, and the procedure can be seen as a model compression technique. Originally, the distillation procedure achieves the knowledge transfer by training the student network using the original training targets, denoted as ground-truth targets, as well as a softened distribution of logits from the (already trained and fixed) teacher network.[2] Since the popularization of knowledge distillation by Hinton et al. (2015), the idea of knowledge distillation has been extended to a variety of settings.[3] This paper will focus on the special case where the teacher and student are of identical architecture, called self-distillation, and where the aim is to improve predictive performance, rather than compressing the model.

The idea of self-distillation is to use outputs from a trained model together with the original targets as new targets for retraining the same model from scratch. We refer to this as one step of self-distillation, and one can iterate this procedure for multiple distillation steps (see Figure 1). Empirically, it has been shown that this procedure often generalizes better than the model trained merely on the

---

[1] Often knowledge distillation is also referred to under the name *Teacher-Student learning*.

[2] We will refer to the weighted outputs of the penultimate layer, i.e. pre-activation of the last layer, as logits.

[3] See Section 2 for a brief overview, or see Wang and Yoon (2020) for a more exhaustive survey

35th Conference on Neural Information Processing Systems (NeurIPS 2021).

original targets, and achieves higher predictive performance on validation data, despite no additional information being provided during training (Furlanello et al., 2018; Ahn et al., 2019; Yang et al., 2018).

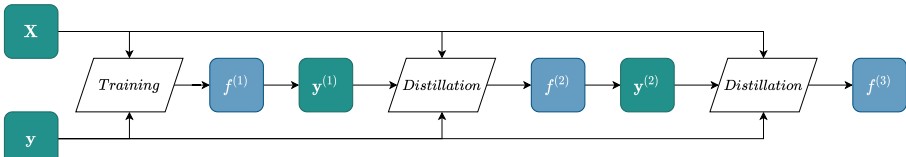

Figure 1: Illustration of self-distillation for two steps after the initial training, where we use the notation $f^{(\tau)} = f(\cdot, \hat{\boldsymbol{\beta}}^{(\tau)})$. See Section 3 for details.

Modern deep neural networks are often trained in the over-parameterized regime, where the amount of trainable parameters highly exceed the amount of training samples. Under simple first-order methods such as gradient descent, such large networks can fit any target, but in order to generalize well, such overfitting is usually undesirable (Zhang et al., 2017; Nakkiran et al., 2020). Thus, some type of regularization is typically imposed during training, in order to avoid overfitting. A common choice is to add an $\ell_2$-regularization[4] term to our objective function, which has been shown to perform comparably to early-stopping gradient descent training (Yao et al., 2007). However, in the theoretical study of the over-parameterized regime, regularization is often overlooked, but recent results have shown a connection between wide neural networks and kernel ridge regression through the Neural Tangent Kernel (NTK) (Lee et al., 2019, 2020; Hu et al., 2019). We briefly elaborate on this connection in Section D, which motivates our problem setup and connection to deep learning in Section 5.

**Our Contributions** Through a theoretical analysis we show that

- the solution at any distillation step can easily be calculated as a function of the initial fit, and infinitely many steps of self-distillation (with fixed distillation weight) correspond to solving the usual kernel ridge regression problem with a specific amplified regularization parameter when the distillation weight is non-zero,

- for fixed distillation weights, self-distillation amplifies the regularization at each distillation step, and the ground-truth targets dampen the sparsification and regularization of the self-distilled solutions, ensuring non-zero solutions for any number of distillation steps,

- the optimal distillation weight has a closed form solution for kernel ridge regression, and can be estimated efficiently for neural networks compared to a grid search.

Proofs of all our results can be found in Supplementary Material A, and code to reproduce our illustrative example in Section 4.5. Experimental results in Section B can be found at github.com/Kennethborup/self_distillation.

## 2 Related Work

The idea of knowledge distillation dates back to Bucila et al. (2006), and was later brought to the deep learning setting by Ba and Caruana (2014) and more recently popularized by Hinton et al. (2015) in the context of compressing neural networks. Since the original formulation, various extensions have been proposed. Some approaches focus on matching the teacher and student models on statistics other than the distribution of the logits, such as intermediate representations (Romero et al., 2015), spacial attention maps (Zagoruyko and Komodakis, 2019), Jacobians (Srinivas and Fleuret, 2018), Gram matrices (Yim et al., 2017), or relational information between teacher outputs (Park et al., 2019). Other extensions focus on developing the transfer procedure, such as self-distillation (Furlanello et al., 2018), data-free distillation (Lopes et al., 2017; Nayak et al., 2019; Micaelli and Storkey, 2019; Chen et al., 2019; Fang et al., 2019), data distillation (Radosavovic et al., 2018), residual knowledge distillation (Gao et al., 2020), online distillation (Anil et al., 2018) or contrastive distillation (Ahn et al., 2019; Tian et al., 2020a).

---

[4]With slight differences, $\ell_2$ regularization is often referred to as weight decay and ridge regularization in deep learning and statistical learning literature, respectively. See e.g. Loshchilov and Hutter (2019) for details.

The practical benefits of knowledge distillation have been proven countless of times in a variety of settings, but the theoretical justification for knowledge distillation is still highly absent. Hinton et al. (2015) conjecture that the success of knowledge distillation should be attributed to the transfer of *dark knowledge* (e.g. inter-class relationships revealed in the soft labels). Müller et al. (2019); Tang et al. (2020) support this conjecture, and argue that knowledge distillation is similar to performing adaptive label smoothing weighted by the teacher's confidence in the predictions. Dong et al. (2019) show the importance of early stopping when training over-parameterized neural networks for distillation purposes by arguing that neural networks tend to fit informative and simple patterns faster than noisy signals, and knowledge distillation utilizes these simple patterns for knowledge transfer. Abnar et al. (2020) empirically investigate how knowledge distillation can transfer inductive biases between student and teacher models, and Gotmare et al. (2019) empirically shows how the dark knowledge shared by the teacher mainly is disbursed to some of the deepest layers of the teacher.

To the best of our knowledge, few papers investigate knowledge distillation from a rigorous theoretical point of view, and those that do, do so with strong assumptions on the setting. Phuong and Lampert (2019) ignore the ground-truth targets during distillation and furthermore assume linear models. Mobahi et al. (2020) investigate self-distillation in a Hilbert space setting with kernel ridge regression models where the teacher is trained on the ground-truth targets, and the student (and subsequent iterations) is only trained on the predictions from the teacher without access to the ground-truth targets. They show that self-distillation progressively limits the number of basis functions used to represent the solutions, thus eventually causing the solutions to underfit. In this paper, we build on the theoretical results of Mobahi et al. (2020), but we include the weighted ground-truth targets in the self-distillation procedure, where we allow the weight to depend on the self-distillation step, and show how this drastically affects the behavior and effect of self-distillation.[5]

# 3  Problem Setup

**Notation**  Vectors and matrices are denoted by bold-faced letters; vectors are column vectors by default, and for a vector $\mathbf{a}$ let $[\mathbf{a}]_i$ be the $i$-th entry and for a matrix $\mathbf{A}$ let $[\mathbf{A}]_{i,j}$ be the $(i,j)$-th entry. Let $\mathbf{I}_n$ denote the identity matrix of dimension $n$, $[k] = \{1, 2, \ldots, k\}$, and let $\|\cdot\|_2$ and $\|\cdot\|_F$ denote the $\ell_2$-norm and the Frobenius norm, respectively. Finally, for a function $h : \mathbb{R}^n \to \mathbb{R}^d$ and $\mathbf{X} \in \mathbb{R}^{m \times n}$, we denote by $h(\mathbf{X})$ the $\mathbb{R}^{m \times d}$ matrix of outcomes, where the $i$'th row of $h(\mathbf{X})$ is the function applied to the $i$'th row of $\mathbf{X}$, i.e. $[h(\mathbf{X})]_{i,\cdot} = h(\mathbf{x}_i)$.

Consider the training dataset $\mathcal{D} \subseteq \mathbb{R}^d \times \mathbb{R}$, and let $\mathcal{X} = \{\mathbf{x} \mid (\mathbf{x}, y) \in \mathcal{D}\}$ and $\mathcal{Y} = \{y \mid (\mathbf{x}, y) \in \mathcal{D}\}$ denote the inputs and targets, respectively. Let $\mathbf{X} = [\mathbf{x}_i]_{i \in [n]} \in \mathbb{R}^{n \times d}$ be the matrix of inputs, $\mathbf{y} = [y_i]_{i \in [n]}$ the vector of targets, and $\tilde{\mathbf{X}} \in \mathbb{R}^{m \times d}, \tilde{\mathbf{y}} \in \mathbb{R}^m$ be the matrix and vector of validation inputs and targets, respectively. Given a feature map $\varphi : \mathbb{R}^d \to \mathcal{V}$, where $\mathcal{V}$ has dimension $D$, we denote by $\mathbf{K} = \kappa(\mathbf{X}, \mathbf{X}) = [\kappa(\mathbf{x}_i, \mathbf{x}_j)]_{i,j=1}^n \in \mathbb{R}^{n \times n}$, where $\kappa(\mathbf{x}_i, \mathbf{x}_j) = \langle \varphi(\mathbf{x}_i), \varphi(\mathbf{x}_j) \rangle$, the symmetric kernel (Gram) matrix associated with the feature map $\varphi$.[6]

## 3.1  Self-Distillation of Kernel Ridge Regressions

In order to avoid overfitting our training data, we will impose a regularization term on our weights, and thus investigate the kernel ridge regression functions $f \in \mathcal{F}$ mapping $f : \mathcal{X} \to \mathcal{Y}$, to construct a solution which best approximates the true underlying data generating map and generalize well to new unseen data from this underlying map. We consider self-distillation in the kernel ridge regression setup; i.e. consider the (self-distillation) objective function

$$\mathcal{L}^{\text{distill}}(f(\mathbf{X}, \boldsymbol{\beta}), \mathbf{y}_1, \mathbf{y}_2) = \frac{\alpha}{2} \|f(\mathbf{X}, \boldsymbol{\beta}) - \mathbf{y}_1\|_2^2 + \frac{1 - \alpha}{2} \|f(\mathbf{X}, \boldsymbol{\beta}) - \mathbf{y}_2\|_2^2 + \frac{\lambda}{2} \|\boldsymbol{\beta}\|_2^2, \quad (1)$$

where $\alpha \in [0, 1], \lambda > 0, \mathbf{y}_1, \mathbf{y}_2 \in \mathbb{R}^n$ and $f(\mathbf{X}, \boldsymbol{\beta}) = \varphi(\mathbf{X})\boldsymbol{\beta}$. The objective in (1) is a weighted sum of two Mean Square Error (MSE) objective functions with different targets[7] and an $\ell_2$-regularization

---

[5]In Supplementary Material E we relate our problem setup to Mobahi et al. (2020) and extend some of our results to a constrained optimization setting with a regularization functional in Hilbert space.

[6]Since the kernel trick makes the predictions depend only on inner products in the feature space, it is not a restriction if $D$ is infinite. However, for ease of exposition we assume $D$ is finite.

[7]It is straightforward to verify that minimizing (1) and the classic MSE objective with a weighted target, i.e. $\tilde{\mathcal{L}}^{\text{distill}}(f(\mathbf{X}, \boldsymbol{\beta}), \mathbf{y}_1, \mathbf{y}_2) = \frac{1}{2} \|f(\mathbf{X}, \boldsymbol{\beta}) - (\alpha\mathbf{y}_1 + (1 - \alpha)\mathbf{y}_2)\|_2^2 + \frac{\lambda}{2} \|\boldsymbol{\beta}\|_2^2$, are equivalent and that the objective functions are equal up to the additive constant $\alpha(\alpha - 1) \|\mathbf{y}_1 - \mathbf{y}_2\|_2^2$.

on the model weights. Minimization of (1) w.r.t. $\boldsymbol{\beta}$ is straightforward and yields the minimizer

$$\hat{\boldsymbol{\beta}} \stackrel{\text{def}}{=} \underset{\beta}{\arg\min}\, \mathcal{L}^{\text{distill}}(f(\mathbf{X}, \boldsymbol{\beta}), \mathbf{y}_1, \mathbf{y}_2) = \varphi(\mathbf{X})^{\intercal}\left(\mathbf{K} + \lambda \mathbf{I}_n\right)^{-1}(\alpha \mathbf{y}_1 + (1 - \alpha)\mathbf{y}_2) \quad (2)$$

by Woodbury's matrix identity and definition of $\mathbf{K}$. This solution can also be seen as a direct application of the Representer Theorem (Schölkopf et al., 2001). Let $\mathbf{y}^{(0)} \stackrel{\text{def}}{=} \mathbf{y}$, i.e. the original targets, and recursively define for the steps $\tau \geq 1$,

$$\hat{\boldsymbol{\beta}}^{(\tau)} \stackrel{\text{def}}{=} \underset{\boldsymbol{\beta}}{\arg\min}\, \mathcal{L}^{\text{distill}}(f(\mathbf{X}, \boldsymbol{\beta}), \mathbf{y}, \mathbf{y}^{(\tau-1)}) \quad (3)$$

$$= \varphi(\mathbf{X})^{\intercal}\left(\mathbf{K} + \lambda \mathbf{I}_n\right)^{-1}\left(\alpha^{(\tau)}\mathbf{y} + (1 - \alpha^{(\tau)})\mathbf{y}^{(\tau-1)}\right),$$

$$f(\mathbf{x}, \hat{\boldsymbol{\beta}}^{(\tau)}) \stackrel{\text{def}}{=} \varphi(\mathbf{x})^{\intercal}\hat{\boldsymbol{\beta}}^{(\tau)} \quad (4)$$

$$= \kappa(\mathbf{x}, \mathbf{X})^{\intercal}\left(\mathbf{K} + \lambda \mathbf{I}_n\right)^{-1}\left(\alpha^{(\tau)}\mathbf{y} + (1 - \alpha^{(\tau)})\mathbf{y}^{(\tau-1)}\right),$$

$$\mathbf{y}^{(\tau)} \stackrel{\text{def}}{=} f(\mathbf{X}, \hat{\boldsymbol{\beta}}^{(\tau)}), \quad (5)$$

for fixed $\alpha^{(\tau)} \in [0, 1]$. Notice, the initial step ($\tau = 1$) corresponds to standard training by definition and as such is independent of $\alpha^{(1)}$. Self-distillation treats the weighted average of the predictions, $\mathbf{y}^{(1)}$, from this initial model on $\mathbf{X}$, and the ground-truth targets, $\mathbf{y}$, as targets. This procedure is repeated as defined in (3)-(5) and we obtain the self-distillation procedure as illustrated in Figure 1. Note, the special cases $\alpha^{(\tau)} = 0$ and $\alpha^{(\tau)} = 1$ correspond to merely training on the predictions from the previous step, and only training on the original targets, respectively. Thus, $\alpha^{(\tau)} = 1$ is usually not of interest, as the solution is equal to a classical kernel ridge regression, and self-distillation plays no role in this scenario. We will often consider the special case of equal weights, $\alpha^{(2)} = \cdots = \alpha^{(\tau)} = \alpha$, and if $\alpha = 0$ this corresponds to the setting investigated in Mobahi et al. (2020) in a slightly different setup. Thus, some of the following results can be seen as a generalization of Mobahi et al. (2020) to step-wise and non-zero $\alpha$.

## 4 Main Results

In this section we present our main results for finitely and infinitely many distillation steps along with a closed form solution for the optimal $\alpha^{(\tau)}$ as well as an illustrative example highlighting the effect of the chosen sequence of $(\alpha^{(t)})$ on the solutions.

### 4.1 Finite Self-Distillation Steps

Our first result, which follows from straightforward computations, states that the predictions obtained after any finite number of distillation steps can be expressed directly as a function of $\mathbf{y}$ and the kernel matrix $\mathbf{K}$ calculated at the initial fit ($\tau = 1$).

**Theorem 4.1.** *Let* $\mathbf{y}^{(\tau)}, \hat{\boldsymbol{\beta}}^{(\tau)}$, *and* $f(\cdot, \hat{\boldsymbol{\beta}}^{(\tau)})$ *be defined as above. Fix* $\alpha^{(2)}, \ldots, \alpha^{(\tau)} \in [0, 1)$, *and let* $\eta(i, \tau) \stackrel{\text{def}}{=} \prod_{j=i}^{\tau}\left(1 - \alpha^{(j)}\right)$, *then for* $\tau \geq 1$, *we have that*

$$\mathbf{y}^{(\tau)} = \left(\sum_{i=2}^{\tau} \alpha^{(i)}\eta(i+1, \tau)\left(\mathbf{K}\left(\mathbf{K} + \lambda \mathbf{I}_n\right)^{-1}\right)^{\tau-i+1} + \eta(2, \tau)\left(\mathbf{K}\left(\mathbf{K} + \lambda \mathbf{I}_n\right)^{-1}\right)^{\tau}\right)\mathbf{y}, \quad (6)$$

$$f(\mathbf{x}, \hat{\boldsymbol{\beta}}^{(\tau)}) = \alpha^{(\tau)}f(\mathbf{x}, \hat{\boldsymbol{\beta}}^{(1)}) + (1 - \alpha^{(\tau)})f(\mathbf{x}, \hat{\boldsymbol{\beta}}^{(\tau)}_{\alpha=0}) \quad (7)$$

*for any* $\mathbf{x} \in \mathbb{R}^d$, *where* $\hat{\boldsymbol{\beta}}^{(\tau)}_{\alpha=0}$ *is the minimizer in* (3) *with* $\alpha^{(\tau)} = 0$.

Since (6) and (7) are expressed only in terms of $\mathbf{K}$, $(\mathbf{K} + \lambda \mathbf{I}_n)^{-1}$, $\kappa(\mathbf{x}, \mathbf{X})$, and $\mathbf{y}$ we are able to calculate the predictions for the training data as well as for any $\mathbf{x} \in \mathbb{R}^d$ based merely on the initial fit ($\tau = 1$) without the need for any additional fits. Hence, despite the calculations of $\mathbf{K}$, $\kappa(\mathbf{x}, \mathbf{X})$, and especially $(\mathbf{K} + \lambda \mathbf{I}_n)^{-1}$ being (potentially) highly computationally demanding, when obtained, we can calculate any distillation step directly by the equations in Theorem 4.1. Furthermore, predictions at step $\tau$ can be seen as a weighted combination of two classical ridge regression solutions, based on the original targets and the predicted targets from step $\tau - 1$, respectively. However, choosing

appropriate $\alpha^{(t)}$ for $t = 2, \ldots, \tau$ is non-trivial. We explore these dynamics in Section 4.3 and 4.4. First, we use Theorem 4.1 to analyse the regularization that self-distillation progressively impose on the solutions.

## 4.2 Effective Sparsification of Self-Distillation Solutions

We now show that we can represent the solutions as a weighted sum of basis functions, and that this basis sparsifies when we increase $\tau$, but also that the amount of sparsification depends on the choice of $\alpha$. A similar sparsification result for the special case of fixed $\alpha^{(\tau)} = 0$ for $\tau \geq 1$ was proved in Mobahi et al. (2020), and in particular, our (13) generalizes equation (47) in their paper.

Using the spectral decomposition of the symmetric matrix $\mathbf{K}$ we write $\mathbf{K} = \mathbf{VDV}^\intercal$, where $\mathbf{V} \in \mathbb{R}^{n \times n}$ is an orthogonal matrix with the eigenvectors of $\mathbf{K}$ as rows and $\mathbf{D} \in \mathbb{R}^{n \times n}$ is a non-negative diagonal matrix with the associated eigenvalues in the diagonal. Inserting the diagonalization yields

$$\mathbf{K}(\mathbf{K} + \lambda \mathbf{I}_n)^{-1} = \mathbf{VDV}^\intercal (\mathbf{VDV}^\intercal + \lambda \mathbf{I}_n)^{-1} \tag{8}$$

$$= \mathbf{VD}\left(\mathbf{D} + \lambda \mathbf{I}_n\right)^{-1}\mathbf{V}^\intercal, \tag{9}$$

where $\lambda > 0$. By straightforward calculations using (6) and (9) we have

$$\mathbf{y}^{(\tau)} = \mathbf{VB}^{(\tau)}\mathbf{V}^\intercal \mathbf{y}, \quad \text{where} \tag{10}$$

$$\mathbf{B}^{(\tau)} \stackrel{\text{def}}{=} \sum_{i=2}^{\tau} \alpha^{(i)} \eta(i+1, \tau) \mathbf{A}^{\tau-i+1} + \eta(2, \tau)\mathbf{A}^\tau, \quad \text{and} \quad \mathbf{A} \stackrel{\text{def}}{=} \mathbf{D}(\mathbf{D} + \lambda \mathbf{I}_n)^{-1}, \tag{11}$$

and $\mathbf{A}, \mathbf{B}^{(\tau)} \in \mathbb{R}^{n \times n}$ are diagonal matrices for any $\tau$. Furthermore, by (10) the only part of the solution depending on $\tau$ is the diagonal matrix, $\mathbf{B}^{(\tau)}$, and in the following we show how $\mathbf{B}^{(\tau)}$ determines the effective sparsification of the solution $f(\cdot, \hat{\boldsymbol{\beta}}^{(\tau)})$.

**Lemma 4.2.** *Let $\mathbf{B}^{(\tau)}$, and $\mathbf{A}$ be defined as above, and let $\mathbf{B}^{(0)} \stackrel{\text{def}}{=} \mathbf{I}_n$. Then we can express $\mathbf{B}^{(\tau)}$ recursively as*

$$\mathbf{B}^{(\tau)} = \mathbf{A}\left((1 - \alpha^{(\tau)})\mathbf{B}^{(\tau-1)} + \alpha^{(\tau)}\mathbf{I}_n\right), \tag{12}$$

*and $[\mathbf{B}^{(\tau)}]_{k,k} \in [0, 1]$ is (strictly) decreasing in $\tau$ for all $k \in [n]$ and $\tau \geq 1$ if $\alpha^{(2)} = \cdots = \alpha^{(\tau)} = \alpha$.*

Similarly to (10), if we use Lemma 4.2 and Theorem 4.1, we can show that for any $\mathbf{x} \in \mathbb{R}^p$

$$f(\mathbf{x}, \hat{\boldsymbol{\beta}}^{(\tau)}) = \kappa(\mathbf{x}, \mathbf{X})^\intercal \mathbf{VD}^{-1}\mathbf{B}^{(\tau)}\mathbf{V}^\intercal \mathbf{y}$$

$$= \mathbf{p}(\mathbf{x})^\intercal \mathbf{B}^{(\tau)}\mathbf{z}, \quad \text{where} \tag{13}$$

$$\mathbf{p}(\mathbf{x}) \stackrel{\text{def}}{=} \mathbf{D}^{-1}\mathbf{V}^\intercal \kappa(\mathbf{x}, \mathbf{X}), \quad \text{and} \quad \mathbf{z} \stackrel{\text{def}}{=} \mathbf{V}^\intercal \mathbf{y}.$$

Thus, the solution $f(\cdot, \hat{\boldsymbol{\beta}}^{(\tau)})$ can be represented as a weighted sum of some basis functions, where the basis functions are the components of the orthogonally transformed and scaled basis $\mathbf{p}(\mathbf{x})$, and $\mathbf{z}$ is an orthogonally transformed vector of targets.

Now assume $\alpha^{(2)} = \cdots = \alpha^{(\tau)} = \alpha$ for any $\tau \geq 2$ for the remaining of this section. In the following we show how the behaviour of $\mathbf{B}^{(\tau)}$, and in turn also the behaviour of $f(\cdot, \hat{\boldsymbol{\beta}}^{(\tau)})$, with $\tau$ is dependent on the choice of $\alpha$. Lemma 4.2 not only provides a recursive formula for $\mathbf{B}^{(\tau)}$, but also shows that each diagonal element of $\mathbf{B}^{(\tau)}$ is in $[0, 1]$ and is strictly decreasing in $\tau$, which in turn implies that the self-distillation procedure progressively shrinks the coefficients of the basis functions. Using Lemma 4.2 we can now show, that not only does $\mathbf{B}^{(\tau)}$ decrease in $\tau$, smaller elements of $\mathbf{B}^{(\tau)}$ shrink faster than larger elements for $\alpha = 0$, as we elaborate on below the theorem.

**Theorem 4.3.** *For any pair of diagonals of $\mathbf{D}$, i.e. $d_k$ and $d_j$, where $d_k > d_j$, we have for all $\tau \geq 1$,*

$$\frac{[\mathbf{B}^{(\tau)}]_{k,k}}{[\mathbf{B}^{(\tau)}]_{j,j}} = \begin{cases} \frac{1 + \frac{\lambda}{d_j}}{1 + \frac{\lambda}{d_k}}, & \text{for } \alpha = 1, \\ \left(\frac{1 + \frac{\lambda}{d_j}}{1 + \frac{\lambda}{d_k}}\right)^\tau, & \text{for } \alpha = 0, \end{cases} \tag{14}$$

*and if we let* $\mathrm{sgn}(\cdot)$ *denote the sign function*[8], *then for* $\alpha \in (0,1)$ *we have that*

$$\mathrm{sgn}\left(\frac{[\mathbf{B}^{(\tau)}]_{k,k}}{[\mathbf{B}^{(\tau)}]_{j,j}} - \frac{[\mathbf{B}^{(\tau-1)}]_{k,k}}{[\mathbf{B}^{(\tau-1)}]_{j,j}}\right)$$

$$= \mathrm{sgn}\left(\left(\left(\frac{[\mathbf{B}^{(\tau-1)}]_{k,k}}{[\mathbf{B}^{(\tau-1)}]_{j,j}} - \frac{[\mathbf{A}]_{k,k}}{[\mathbf{A}]_{j,j}}\right)\frac{[\mathbf{A}]_{j,j}}{[\mathbf{B}^{(\tau-1)}]_{k,k}([\mathbf{A}]_{k,k} - [\mathbf{A}]_{j,j})} + 1\right)^{-1} - \alpha\right). \quad (15)$$

If we consider a pair of diagonals of $\mathbf{D}$, where $d_k > d_j$, then for $\alpha = 0$, the fraction $\frac{[\mathbf{B}^{(\tau)}]_{k,k}}{[\mathbf{B}^{(\tau)}]_{j,j}}$ is strictly increasing in $\tau$, due to the r.h.s. of (14) inside the parenthesis being strictly larger than 1. Hence, the diagonals corresponding to smaller eigenvalues shrink faster than the larger ones as $\tau$ increases. However, for $\alpha \in (0,1)$ we can not ensure this behaviour, but at step $\tau$ we are able to predict the behaviour at step $\tau + 1$, by using (15). Thus, when we include the ground-truth targets in our distillation procedure we do not consistently increase the regularization with each distillation step, but can potentially obtain a solution which does not sparsify any further. We now turn our attention to the question of how to pick the $\alpha^{(\tau)}$'s in an optimal manner, and find that it can be done if we relax the condition that the weights are restricted to the interval $[0, 1]$.

### 4.3 Closed Form Optimal Weighting Parameter

Recall, $\tilde{\mathbf{X}} \in \mathbb{R}^{m \times d}$ is the matrix of validation inputs and $\tilde{\mathbf{y}} \in \mathbb{R}^m$ the vector of validation targets. If we allow $\alpha^{(\tau)} \in \mathbb{R}$, we can find an *optimal* $\alpha^{(\tau)}$ (which is a non-trivial function of $\lambda$) at each step $\tau$, denoted by $\alpha^{\star(\tau)}$.[9] Here, *optimal* denotes the value for which the validation MSE is minimized. Note, $\alpha^{\star(\tau)}$ is optimal for a single distillation step, but not necessarily so for multiple distillation steps, however we may consider $\alpha^{\star(\tau)}$ a greedy estimate of the optimal value across multiple steps.

**Theorem 4.4.** *Fix* $\tau \geq 2$, $\lambda > 0$ *and* $\alpha^{(2)}, \ldots, \alpha^{(\tau-1)} \in \mathbb{R}$, *then*

$$\alpha^{\star(\tau)} = \underset{\alpha^{(\tau)} \in \mathbb{R}}{\mathrm{argmin}} \left\|\tilde{\mathbf{y}} - f(\tilde{\mathbf{X}}, \hat{\boldsymbol{\beta}}^{(\tau)})\right\|_2^2 = 1 - \frac{\left(\tilde{\mathbf{y}}_{\alpha=0}^{(\tau)} - \tilde{\mathbf{y}}^{(1)}\right)^{\mathsf{T}}\left(\tilde{\mathbf{y}} - \tilde{\mathbf{y}}^{(1)}\right)}{\left\|\tilde{\mathbf{y}}_{\alpha=0}^{(\tau)} - \tilde{\mathbf{y}}^{(1)}\right\|_2^2} \quad (16)$$

*where* $\tilde{\mathbf{y}}^{(1)} = f(\tilde{\mathbf{X}}, \hat{\boldsymbol{\beta}}^{(1)})$, *and* $\tilde{\mathbf{y}}_{\alpha=0}^{(\tau)} = f(\tilde{\mathbf{X}}, \hat{\boldsymbol{\beta}}_{\alpha=0}^{(\tau)})$.

Since neither $\tilde{\mathbf{y}}^{(1)}$ nor $\tilde{\mathbf{y}}_{\alpha=0}^{(\tau)}$ depend on the choice of $\alpha^{(\tau)}$, we can calculate $\alpha^{\star(\tau)}$ recursively as presented in Algorithm 1, where $\alpha^{\star(\tau)}$ has the closed form in (16). In combination with the diagonalization results of Section 4.2 we can efficiently calculate the solutions. This should be compared to performing grid-search for $\alpha$ with $g$ equidistant values on $[0, 1]$ in order to approximate the optimal $\alpha$, which requires $g(\tau - 1) + 1$ model fits if one uses the same $\alpha$ for each sequence of $\tau \geq 2$ steps ($g^{\tau-1}$ if $\alpha$ is not fixed across distillation steps). However, by Algorithm 1 it is sufficient to perform $2(\tau - 1) + 1$ model fits, and obtain the exact optimal value at each step instead of an approximated value. In Section 5 we apply Algorithm 1 to approximate $\alpha^{\star(\tau)}$ in a deep learning setting.

### 4.4 Infinite Number of Self-Distillation Steps

We now prove that if we were to perform an infinite number of distillations steps ($\tau \to \infty$) with a fixed $\alpha$ (i.e. $\alpha^{(2)} = \cdots = \alpha^{(\tau)} = \alpha$) the solution would solve the classical kernel ridge regression problem, with an amplified regularization parameter (by $\alpha^{-1}$) if $\alpha > 0$. Observe that, when $\alpha = 0$ and $\tau \to \infty$, (6) and (7) yield that the predictions $\mathbf{y}^{(\infty)}$ and $f(\mathbf{x}, \hat{\boldsymbol{\beta}}^{(\infty)})$ collapse to the zero-solution for any $\mathbf{x} \in \mathbb{R}^p$ as expected from Mobahi et al. (2020).

---

[8]Note, we use the definition of $\mathrm{sgn}(\cdot)$ where $\mathrm{sgn}(0) \overset{\mathrm{def}}{=} 0$.

[9]If $\alpha^{\star(\tau)} \notin [0, 1]$, the sign of either the first or second term of (1) becomes negative, indicating either too strong or weak regularization of the previous distillation step, and one might fear this affects distillation performance. However, simply clipping of $\alpha^{\star(\tau)}$ to be in $[0, 1]$ alleviates this, at the cost of requiring a larger $\tau$.

**Theorem 4.5.** *Let* $\mathbf{y}^{(\tau)}, \hat{\boldsymbol{\beta}}^{(\tau)}$, *and* $f(\cdot, \hat{\boldsymbol{\beta}}^{(\tau)})$ *be defined as above, and* $\alpha \in (0, 1]$, *then the following limits hold*

$$\mathbf{y}^{(\infty)} \stackrel{\text{def}}{=} \lim_{\tau \to \infty} \mathbf{y}^{(\tau)} = \mathbf{K} \left( \mathbf{K} + \frac{\lambda}{\alpha} \mathbf{I}_n \right)^{-1} \mathbf{y} \tag{17}$$

$$f(\mathbf{x}, \hat{\boldsymbol{\beta}}^{(\infty)}) \stackrel{\text{def}}{=} \lim_{\tau \to \infty} f(\mathbf{x}, \hat{\boldsymbol{\beta}}^{(\tau)}) = \alpha f(\mathbf{x}, \hat{\boldsymbol{\beta}}^{(1)}) + (1 - \alpha) f(\mathbf{x}, \hat{\boldsymbol{\gamma}}^{(\infty)})$$

*where* (17) *corresponds to* classical *kernel ridge regression with amplified regularization parameter* $\lambda/\alpha$, *and we let* $\hat{\boldsymbol{\gamma}}^{(\infty)}$ *denote the kernel ridge regression weights associated with solving another kernel ridge regression on the targets* $\mathbf{y}^{(\infty)}$ *with regularization parameter* $\lambda$. *Furthermore, the convergence* $\lim_{\tau \to \infty} \mathbf{y}^{(\tau)}$ *is of linear rate.*

If $\alpha > 0$, then by (9) and Theorem 4.5, we have that $\mathbf{y}^{(\infty)} = \sum_{j=1}^{p} \mathbf{v}_j \frac{d_j}{d_j + \frac{\lambda}{\alpha}} \mathbf{v}_j^{\mathsf{T}} \mathbf{y}$ and we shrink the eigenvectors with smallest eigenvalues, corresponding to the directions with least variance, the most. Furthermore, if $\alpha > 0$ the limiting solution is a non-zero kernel ridge regression with regularization parameter $\lambda/\alpha \geq \lambda$, causing the eigenvectors associated with the smallest eigenvalues to shrink even more than in the original solution.

---

**Algorithm 1:** Calculate $\hat{\boldsymbol{\beta}}^{(\tau)}$ and $\alpha^{\star(\tau)}$ for $\tau \geq 2$.

Calculate $\hat{\boldsymbol{\beta}}^{(1)}$ from (3) (with any $\alpha^{(1)}$);
Calculate $\tilde{\mathbf{y}}^{(1)} = f(\tilde{\mathbf{X}}, \hat{\boldsymbol{\beta}}^{(1)})$;
**for** $t = 2$ **to** $\tau$ **do**
    Calculate $\hat{\boldsymbol{\beta}}_{\alpha=0}^{(t)}$ from (3) and $\tilde{\mathbf{y}}_{\alpha=0}^{(t)} = f(\tilde{\mathbf{X}}, \hat{\boldsymbol{\beta}}_{\alpha=0}^{(t)})$;
    Solve:
$$\alpha^{\star(t)} = \operatorname*{argmin}_{\alpha \in \mathbb{R}} \left\| \tilde{\mathbf{y}} - \left( \alpha \tilde{\mathbf{y}}^{(1)} + (1 - \alpha) \tilde{\mathbf{y}}_{\alpha=0}^{(t)} \right) \right\|_2^2;$$
    Calculate $\hat{\boldsymbol{\beta}}^{(t)}$ from (3) with $\alpha^{\star(t)}$;
**end**

---

Our results gives a theoretical explanation for why one should treat $\alpha^{(\tau)}$ as an adjustable hyperparameter to fine-tune the amount of regularization that self-distillation impose for a particular problem, and that it can be chosen in an optimal way for kernel ridge regression. In the following we provide an illustrative example, and in Section 5 we estimate the optimal weighting parameter for deep learning using an adaptation of Algorithm 1.

### 4.5 Illustrative example

Consider the training dataset $\mathcal{D}$ where $\mathcal{X} = \{0, 0.1, \dots, 0.9, 1\}$ and $\mathcal{Y} = \{\sin(2\pi x) + \varepsilon \mid x \in \mathcal{X}\}$, and $\varepsilon$ is sampled from a zero-mean Gaussian random variable with standard deviation $0.5$. Let $\varphi$ be the Radial Basis Function kernel, i.e. $\kappa(\mathbf{x}_i, \mathbf{x}_j) = e^{-\gamma \|\mathbf{x}_i - \mathbf{x}_j\|_2^2}$, where we choose $\gamma = \frac{1}{80}$, and let $\lambda = 0.2$ and consider the three cases; *(a)* $\alpha = 0$, *(b)* $\alpha = 0.25$, and *(c)* step-wise optimal $\alpha^{\star(\tau)}$.

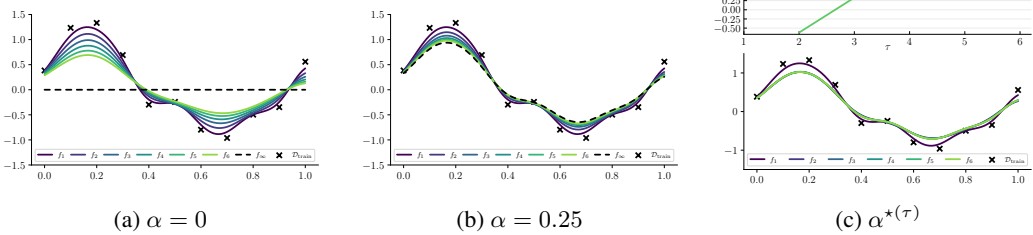

(a) $\alpha = 0$        (b) $\alpha = 0.25$        (c) $\alpha^{\star(\tau)}$

Figure 2: Six steps of self-distillation with (a) zero limiting solution (dashed), (b) non-zero limiting solution (dashed), and (c) optimal step-wise $\alpha^{\star(\tau)}$. Training examples are represented with $\times$ and in (c) we also plot $\alpha^{\star(\tau)}$ with $\tau$.

As illustrated in Figure 2a for case *(a)*, the regularization imposed by self-distillation initially improves the quality of the solution, but eventually overregularize and the solutions underfit the data, and will eventually converge to the zero-solution (see supplementary materials for the loss values). Using $\alpha > 0$ (see Figure 2b), and more specifically $\alpha = 0.25$, reduce the imposed regularization and increases the stability of the distillation procedure; i.e. the solutions differ much less between each distillation step. This allows for a more dense exploration of solutions during iterated distillation steps, where increasing $\alpha$ reduces the difference between solutions from two consecutive steps, but

also reduces the space of possible solutions as the limit, $f(\cdot, \hat{\boldsymbol{\beta}}^{(\infty)})$, approaches the initial solution $f(\cdot, \hat{\boldsymbol{\beta}}^{(1)})$ quickly.[10]. However, choosing the step-wise optimal $\alpha^{\star(\tau)}$ yields minuscule changes to the solution for $\tau > 2$, and a single step of distillation is effectively enough. Furthermore, for $\tau \geq 3$, all $\alpha^{\star(\tau)}$ are approximately equal, and the distillation procedure has reached an equilibrium.[11]

As expected from Lemma 4.2 and Theorem 4.3, Figure 3 verifies that both in case (*a*) and (*b*), the diagonal of $\mathbf{B}^{(\tau)}$ is decreasing in $\tau$ and the diagonal coordinates corresponding to smaller eigenvalues shrink faster than those corresponding to larger eigenvalues in case (*a*). Without loss of generality we can assume $d_1 < d_2 < \cdots < d_n$, and for $k = 1, \ldots n-1$ and any $\tau \geq 1$ define $R_k^{(\tau)} \stackrel{\text{def}}{=} \frac{[\mathbf{B}^{(\tau)}]_{k+1,k+1}}{[\mathbf{B}^{(\tau)}]_{k,k}}$. We expect $R_k^{(\tau)}$ to be strictly increasing in $\tau$ for all $k$ in case (*a*), but for case (*b*) we can make no such guarantee. Both of these properties are verified in Figure 4.

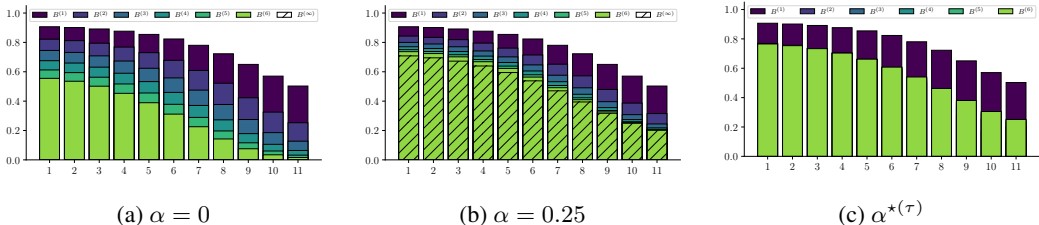

(a) $\alpha = 0$        (b) $\alpha = 0.25$        (c) $\alpha^{\star(\tau)}$

Figure 3: Diagonal of $\mathbf{B}^{(\tau)}$ for $\tau = 1, \ldots, 6$ associated with Figure 2. Note, the plots are overlaid, but since the diagonal of $\mathbf{B}^{(\tau)}$ decrease in $\tau$, all values until convergence are visible. In (a) we expect and observe strictly decreasing values in $\tau$ for all indices, until collapsing at 0, but in (b) and (c) the values converge to a non-zero limit.

Finally, we observe that in case (*a*), the values of $\mathbf{B}^{(\tau)}$ shrink much faster than in case (*b*), and eventually collapse to all zeros, whereas the latter is nearly converged after six iterations. Furthermore, case (*a*) appear to obtain a more sparsified solution, as the smallest coordinates effectively diminishes, which is not true for case (*b*). Furthermore, when directly comparing solutions from both cases with similar quality of fit, the solutions obtained with $\alpha = 0$ usually has smaller coordinates in $\mathbf{B}^{(\tau)}$ than those obtained with larger values of $\alpha$.

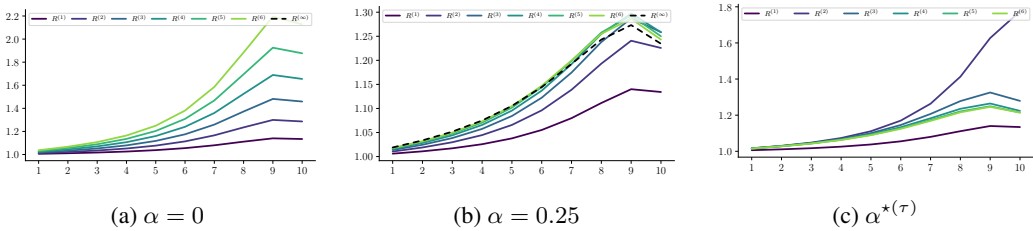

(a) $\alpha = 0$        (b) $\alpha = 0.25$        (c) $\alpha^{\star(\tau)}$

Figure 4: Ratios, $R_k^{(\tau)}$ of the ordered diagonal of $\mathbf{B}^{(\tau)}$ for all $\tau$. In (a) we expect and observe strictly increasing values in $\tau$ for all $k$, but have no such guarantee in (b) or (c). The x-axis corresponds to indices $k = 1, \ldots, n-1$.

## 5 Approximate Optimal Weighting Parameter for Deep Learning

The following experiment aim at empirically evaluating the theoretical analysis above in a simple deep learning setting. In (16) we find $\alpha^{\star(\tau)}$ on closed form when $f(\cdot, \hat{\boldsymbol{\beta}}^{(\tau)})$ is a (self-distilled) kernel ridge regression. No closed form solution can be found for neural networks, but recent results show that (very) wide neural networks can be seen as kernel ridge regression solutions with the neural tangent kernel (Jacot et al., 2018; Arora et al., 2019; Lee et al., 2019, 2020).

---

[10]As expected by Theorem 4.5, we experience a fast convergence to the limit; usually less than 10 iterations are sufficient to converge

[11]If we clip $\alpha^{\star(\tau)}$ to be in $[0, 1]$, the $\alpha^{\star(\tau)}$ converges at $\tau = 4$ rather than $\tau = 3$.

Thus, inspired by (7) we propose to estimate $\alpha^{\star(t)}$ for $t = 2, \ldots, \tau$, denoted by $\hat{\alpha}^{(t)}$, for a neural network trained with self-distillation using an adapted Algorithm 1. Let $f_{\mathrm{nn}}(\cdot, \boldsymbol{\theta}) \in \mathbb{R}^p$ be a neural network with vector of weights $\boldsymbol{\theta}$, and recursively for $\tau \geq 1$ let $\hat{\boldsymbol{\theta}}^{(\tau)}$ be the weights solving

$$\underset{\theta}{\operatorname{argmin}} \frac{\alpha^{(\tau)}}{2} \left\| f_{\mathrm{nn}}(\mathbf{X}, \boldsymbol{\theta}) - \mathbf{Y}^{(1)} \right\|_F^2 + \frac{1 - \alpha^{(\tau)}}{2} \left\| f_{\mathrm{nn}}(\mathbf{X}, \boldsymbol{\theta}) - \mathbf{Y}^{(\tau-1)} \right\|_F^2 + \frac{\lambda}{2} \left\| \boldsymbol{\theta} \right\|_2^2, \quad (18)$$

with $\alpha^{(\tau)} = \hat{\alpha}^{(\tau)}$ and where $\mathbf{Y}^{(\tau)} \in \mathbb{R}^{n \times p}$.[12] Furthermore, let $\hat{\boldsymbol{\theta}}_{\alpha=0}^{(\tau)}$ be the weights associated with minimizing (18) with $\alpha^{(\tau)} = 0$, and $\tilde{\mathbf{Y}}_{\alpha=0}^{(\tau)} \overset{\text{def}}{=} f_{\mathrm{nn}}(\tilde{\mathbf{X}}, \hat{\boldsymbol{\theta}}_{\alpha=0}^{(\tau)})$ as well as $\tilde{\mathbf{Y}}^{(\tau)} \overset{\text{def}}{=} f_{\mathrm{nn}}(\tilde{\mathbf{X}}, \hat{\boldsymbol{\theta}}^{(\tau)})$ be the predictions on the validation input $\tilde{\mathbf{X}}$. Then, following Algorithm 1 with $\|\cdot\|_2$ replaced by $\|\cdot\|_F$, and (18) rather than (1) we can calculate the estimates $\hat{\alpha}^{(t)}$. These estimates yield comparable predictive performance to the best fixed $\alpha^{(\tau)}$ (found with time-consuming grid search), but only require one additional model fit per distillation step; i.e. $2(\tau - 1) + 1$ fits compared to $g(\tau - 1) + 1$ for a grid search over $g$ values. See Figure 5 for results and supplementary material for experimental details.

## 5.1 Experiment

We perform self-distillation with ResNet-50 (He et al., 2016) networks on CIFAR-10 (Krizhevsky and Hinton, 2009), with minor pre-processing and augmentations. The model is initialized randomly at each step[13] and trained according to the above with either estimated optimal parameters, $\hat{\alpha}^{(\tau)}$, or fixed $\alpha$ for all steps. We use the network weights from the last iteration of training at each distillation step for the next step, irrespective of whether a better model occurred earlier in the training. Our models are trained for a fixed 75 epochs and each experiment is repeated with 4 different random seeds over 11 chains of distillation steps, corresponding to $\alpha \in \{0.0, 0.1, \ldots, 0.9\}$ and $\hat{\alpha}^{(\tau)}$, with the first model initialized identically across all chains. The accuracy reported at the $\tau$'th step is based on comparing the training and validation predictions, $\mathbf{Y}^{(\tau)}$ and $f(\tilde{\mathbf{X}}, \hat{\boldsymbol{\beta}}^{(\tau)})$ with the original training and validation targets; $\mathbf{Y}$ and $\tilde{\mathbf{Y}}$.[14]

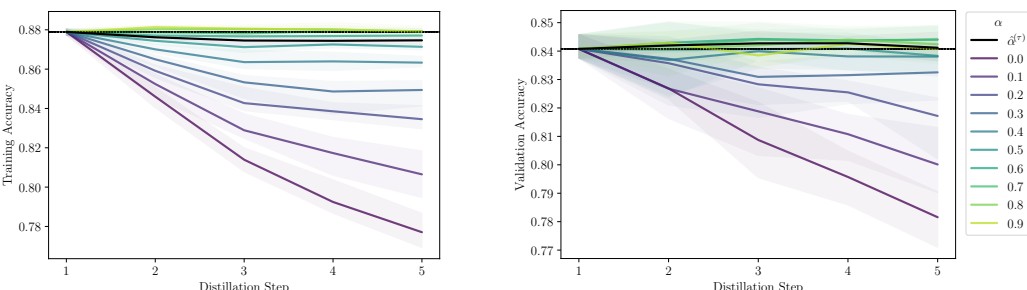

Figure 5: Training and validation accuracy for five distillation steps with ResNet-50 models on CIFAR-10. Comparing fixed $\alpha^{(t)}$ for $t = 2 \ldots, \tau$ and estimating optimal weight with $\hat{\alpha}^{(t)}$ at each step. The experiment is repeated four times and the mean (and max/min in shaded) is reported.

## 6 Conclusion

In this paper, we provided theoretical arguments for the importance of weighting the teacher outputs with the ground-truth targets when performing self-distillation with kernel ridge regressions along with a closed form solution for the optimal weighting parameter. We proved how the solution at any (possibly infinite) distillation step can be calculated directly from the initial distillation step, and that self-distillation for an infinite number of steps corresponds to a classical kernel ridge regression solution with amplified regularization parameter. We showed both empirically and theoretically that

---

[12]We treat class labels as $p$-dimensional one-hot encoded vectors and use norm of the difference between the predicted class probabilities and the one-hot vectors.

[13]Note, we initialize the models equally across all $\alpha$ for one experiment, but alter the seed for initialization between experiments.

[14]The empirical experiments are constrained by the theoretical set-up and performed in a highly simple setting; e.g. using the weighted MSE loss from (18) (see supplementary materials for more details). Therefore, our accuracy measures are to be expected to be lower than for more fine-tuned training setups.

the weighting parameter $\alpha$ determines the amount of regularization imposed by self-distillation, and empirically supported our results in a simple deep learning setting.

### 6.1 Future Research Directions

Interesting directions of future research are on rigorously connecting neural networks and kernel methods in a knowledge distillation setting, extend to other objective functions than MSE as well as including intermediate model statistics in the distillation procedure. Finally, a larger empirical study of the connection between the choice of $\alpha$ and the degree of overfitting is interesting as well.

## Acknowledgments and Disclosure of Funding

We would like to thank GenomeDK and Aarhus University for providing computational resources that contributed to these research results. Furthermore, we would like to thank Daniel Borup and Ragnhild Ø. Laursen for comments and discussion, as well as Google Researcher Hossein Mobahi and Mehrdad Farajtabar (Deepmind) for clarifications on their experimental setup. We also thank the anonymous reviewers of the NeurIPS 2021 conference for their comments. Kenneth Borup is partly financed by Aarhus University Centre for Digitalisation, Big Data and Data Analytics (DIGIT).

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
