# A Proofs

This section includes all proofs referenced in the main part of the paper, along with the associated theorems and lemmas for completeness.

**Theorem A.1.** *Let $\mathbf{y}^{(\tau)}, \hat{\boldsymbol{\beta}}^{(\tau)}$, and $f(\cdot, \hat{\boldsymbol{\beta}}^{(\tau)})$ be defined as above. Fix $\alpha^{(2)}, \ldots, \alpha^{(\tau)} \in [0, 1)$, and let $\eta(i, \tau) \stackrel{\text{def}}{=} \prod_{j=i}^{\tau} (1 - \alpha^{(j)})$, then for $\tau \geq 1$, we have that*

$$\mathbf{y}^{(\tau)} = \left( \sum_{i=2}^{\tau} \alpha^{(i)} \eta(i+1, \tau) \left( \mathbf{K} \left( \mathbf{K} + \lambda \mathbf{I}_n \right)^{-1} \right)^{\tau - i + 1} + \eta(2, \tau) \left( \mathbf{K} \left( \mathbf{K} + \lambda \mathbf{I}_n \right)^{-1} \right)^{\tau} \right) \mathbf{y},$$

$$f(\mathbf{x}, \hat{\boldsymbol{\beta}}^{(\tau)}) = \alpha^{(\tau)} f(\mathbf{x}, \hat{\boldsymbol{\beta}}^{(1)}) + (1 - \alpha^{(\tau)}) f(\mathbf{x}, \hat{\boldsymbol{\beta}}_{\alpha=0}^{(\tau)})$$

*for any $\mathbf{x} \in \mathbb{R}^d$, where $\hat{\boldsymbol{\beta}}_{\alpha=0}^{(\tau)}$ is the minimizer in (3) with $\alpha^{(\tau)} = 0$.*

*Proof.* We prove the theorem by induction, where we let $\tilde{\mathbf{K}} \stackrel{\text{def}}{=} \mathbf{K}(\mathbf{K} + \lambda \mathbf{I}_n)^{-1}$. For $\tau = 1$, the result hold trivially, and thus, assume it hold for $\tau = t$. Since $\boldsymbol{\beta}^{(t+1)} = \varphi(\mathbf{X})^{\mathsf{T}}(\mathbf{K} + \lambda \mathbf{I}_n)^{-1} \left( \alpha^{(t+1)} \mathbf{y} + (1 - \alpha^{(t+1)}) \mathbf{y}^{(t)} \right)$ we have that

$$\mathbf{y}^{(t+1)} = \varphi(\mathbf{X}) \varphi(\mathbf{X})^{\mathsf{T}} (\mathbf{K} + \lambda \mathbf{I}_n)^{-1} \left( \alpha^{(t+1)} \mathbf{y} + (1 - \alpha^{(t+1)}) \mathbf{y}^{(t)} \right)$$

$$= \alpha^{(t+1)} \tilde{\mathbf{K}} \mathbf{y} + (1 - \alpha^{(t+1)}) \tilde{\mathbf{K}} \left( \sum_{i=2}^{t} \alpha^{(i)} \eta(i+1, t) \tilde{\mathbf{K}}^{t-i+1} + \eta(2, t) \tilde{\mathbf{K}}^t \right) \mathbf{y}$$

$$= \alpha^{(t+1)} \tilde{\mathbf{K}} \mathbf{y} + \left( \sum_{i=2}^{t} \alpha^{(i)} \eta(i+1, t+1) \tilde{\mathbf{K}}^{(t+1)-i+1} + \eta(2, t+1) \tilde{\mathbf{K}}^{t+1} \right) \mathbf{y}$$

$$= \left( \sum_{i=2}^{t+1} \alpha^{(i)} \eta(i+1, t+1) \tilde{\mathbf{K}}^{(t+1)-i+1} + \eta(2, t+1) \tilde{\mathbf{K}}^{t+1} \right) \mathbf{y}$$

which finalizes our induction proof for the first part. For the second part, note that it also holds trivially for $\tau = 1$. Thus assume, it holds for $\tau = t$, then by direct manipulations

$$f(\mathbf{x}, \boldsymbol{\beta}^{(t+1)}) = \kappa(\mathbf{x}, \mathbf{X})^{\mathsf{T}} (\mathbf{K} + \lambda \mathbf{I}_n)^{-1} \left( \alpha^{(t+1)} \mathbf{y} + (1 - \alpha^{(t+1)}) \mathbf{y}^{(t)} \right)$$

$$= \alpha^{(t+1)} f(\mathbf{x}, \boldsymbol{\beta}^{(1)}) + (1 - \alpha^{(t+1)}) \kappa(\mathbf{x}, \mathbf{X})^{\mathsf{T}} (\mathbf{K} + \lambda \mathbf{I}_n)^{-1} \mathbf{y}^{(t)}$$

$$= \alpha^{(t+1)} f(\mathbf{x}, \boldsymbol{\beta}^{(1)}) + (1 - \alpha^{(t+1)}) f(\mathbf{x}, \hat{\boldsymbol{\beta}}_{\alpha=0}^{(t+1)}),$$

where we let $\hat{\boldsymbol{\beta}}_{\alpha=0}^{(t+1)}$ denote the minimizer (3) with $\alpha^{(t+1)} = 0$; i.e. minimizing the classical kernel ridge regression problem with targets $\mathbf{y}^{(t)}$. $\qquad \square$

**Lemma A.2.** *Let $\mathbf{B}^{(\tau)}$, and $\mathbf{A}$ be defined as above, and let $\mathbf{B}^{(0)} \stackrel{\text{def}}{=} \mathbf{I}$. Then we can express $\mathbf{B}^{(\tau)}$ recursively as*

$$\mathbf{B}^{(\tau)} = \mathbf{A} \left( (1 - \alpha^{(\tau)}) \mathbf{B}^{(\tau-1)} + \alpha^{(\tau)} \mathbf{I}_n \right),$$

*and $[\mathbf{B}^{(\tau)}]_{k,k} \in [0, 1]$ is (strictly) decreasing in $\tau$ for all $k \in [n]$ and $\tau \geq 1$ if $\alpha^{(2)} = \cdots = \alpha^{(\tau)} = \alpha$.*

*Proof.* The case, $\tau = 1$, is easy to verify, and we assume the claim holds for $\tau = t$. Then note that

$$\mathbf{A} \left( (1 - \alpha^{(t+1)}) \mathbf{B}^{(t)} + \alpha^{(t+1)} \mathbf{I}_n \right) = \sum_{i=2}^{t} \alpha^{(i)} \eta(i+1, t+1) \mathbf{A}^{(t+1)-i+1} + \eta(2, t+1) \mathbf{A}^{t+1} + \alpha^{(t+1)} \mathbf{A}$$

$$= \sum_{i=2}^{t+1} \alpha^{(i)} \eta(i+1, t+1) \mathbf{A}^{(t+1)-i+1} + \eta(2, t+1) \mathbf{A}^{t+1}$$

$$= \mathbf{B}^{(t+1)},$$

finalizing the induction proof. Now, assume $\alpha^{(2)} = \cdots = \alpha^{(\tau)} = \alpha$ and note that for any $k$ and $\tau \geq 1$, then

$$[\mathbf{A}]_k \left( (1-\alpha)[\mathbf{B}^{(\tau-1)}]_{k,k} + \alpha \right) = [\mathbf{B}^{(\tau)}]_{k,k} \leq [\mathbf{B}^{(\tau-1)}]_{k,k} = [\mathbf{A}]_k \left( (1-\alpha)[\mathbf{B}^{(\tau-2)}]_{k,k} + \alpha \right),$$

if and only if $[\mathbf{B}^{(\tau-1)}]_{k,k} \leq [\mathbf{B}^{(\tau-2)}]_{k,k}$, and iteratively, if and only if $[\mathbf{B}^{(1)}]_{k,k} \leq [\mathbf{B}^{(0)}]_{k,k}$. The latter is indeed true, since $\mathbf{B}^{(1)} = \mathbf{A}$, and finally, $\mathbf{A} = \mathbf{I}_n$ if and only if $\lambda = 0$. $\qquad\square$

**Theorem A.3.** *Assume $\alpha^{(2)} = \cdots = \alpha^{(\tau)} = \alpha$. Then, for any pair of diagonals of $\mathbf{D}$, i.e. $d_k$ and $d_j$, where $d_k > d_j$, we have that for all $\tau \geq 1$,*

$$\frac{[\mathbf{B}^{(\tau)}]_{k,k}}{[\mathbf{B}^{(\tau)}]_{j,j}} = \begin{cases} \frac{1+\frac{\lambda}{d_j}}{1+\frac{\lambda}{d_k}}, & \text{for } \alpha = 1, \\[3mm] \left( \frac{1+\frac{\lambda}{d_j}}{1+\frac{\lambda}{d_k}} \right)^{\tau}, & \text{for } \alpha = 0, \end{cases}$$

*and if we let* $\mathrm{sgn}(\cdot)$ *denote the sign function, i.e.*

$$\mathrm{sgn}(x) \overset{\text{def}}{=} \begin{cases} 1 & \text{if } x > 0 \\ 0 & \text{if } x = 0 \,, \\ -1 & \text{if } x < 0 \end{cases}$$

*then for $\alpha \in (0,1)$ we have that*

$$\mathrm{sgn}\left( \frac{[\mathbf{B}^{(\tau)}]_{k,k}}{[\mathbf{B}^{(\tau)}]_{j,j}} - \frac{[\mathbf{B}^{(\tau-1)}]_{k,k}}{[\mathbf{B}^{(\tau-1)}]_{j,j}} \right)$$
$$= \mathrm{sgn}\left( \left( \left( \frac{[\mathbf{B}^{(\tau-1)}]_{k,k}}{[\mathbf{B}^{(\tau-1)}]_{j,j}} - \frac{[\mathbf{A}]_{k,k}}{[\mathbf{A}]_{j,j}} \right) \frac{[\mathbf{A}]_{j,j}}{[\mathbf{B}^{(\tau-1)}]_{k,k}([\mathbf{A}]_{k,k} - [\mathbf{A}]_{j,j})} + 1 \right)^{-1} - \alpha \right).$$

*Proof.* First note that

$$\frac{[\mathbf{A}]_{k,k}}{[\mathbf{A}]_{j,j}} = \frac{\frac{d_k}{d_k+\lambda}}{\frac{d_j}{d_j+\lambda}} = \frac{1+\frac{\lambda}{d_j}}{1+\frac{\lambda}{d_k}},$$

and for $\alpha = 1$, (12) amounts to $\mathbf{B}^{(\tau)} = \mathbf{A}$, which gives the first result. For $\alpha = 0$, (12) amounts to $\mathbf{B}^{(\tau)} = \mathbf{A}^{\tau}$, and the second result follows. For the remainder we denote $[\mathbf{B}^{(\tau-1)}]_{k,k}$ by $\mathbf{B}_k$ and $[\mathbf{A}]_{k,k}$ by $\mathbf{A}_k$ to simplify notation. We investigate the case where both r.h.s. and l.h.s. equals zero. Thus, for $\alpha \in (0,1)$, we observe that if

$$\frac{\mathbf{B}_k}{\mathbf{B}_j} = \frac{\mathbf{A}_k}{\mathbf{A}_j} \frac{(1-\alpha)\mathbf{B}_k + \alpha}{(1-\alpha)\mathbf{B}_j + \alpha} = \frac{\mathbf{A}_k}{\mathbf{A}_j} \frac{\frac{1-\alpha}{\alpha}\mathbf{B}_k + 1}{\frac{1-\alpha}{\alpha}\mathbf{B}_j + 1},$$

then we have that

$$\mathbf{B}_j = \frac{1}{\frac{\mathbf{A}_k}{\mathbf{A}_j}\left( \frac{\frac{1-\alpha}{\alpha}\mathbf{B}_k + 1}{\mathbf{B}_k} \right) - \frac{1-\alpha}{\alpha}} = \frac{\mathbf{B}_k}{\frac{\mathbf{A}_k}{\mathbf{A}_j}\left( \frac{1-\alpha}{\alpha}\mathbf{B}_k + 1 \right) - \frac{1-\alpha}{\alpha}\mathbf{B}_k}$$

$$\frac{1-\alpha}{\alpha}\mathbf{B}_j + 1 = \frac{\frac{1-\alpha}{\alpha}\mathbf{B}_k}{\frac{\mathbf{A}_k}{\mathbf{A}_j}\left( \frac{1-\alpha}{\alpha}\mathbf{B}_k + 1 \right) - \frac{1-\alpha}{\alpha}\mathbf{B}_k} + 1 = \frac{\frac{\mathbf{A}_k}{\mathbf{A}_j}\left( \frac{1-\alpha}{\alpha}\mathbf{B}_k + 1 \right)}{\frac{\mathbf{A}_k}{\mathbf{A}_j}\left( \frac{1-\alpha}{\alpha}\mathbf{B}_k + 1 \right) - \frac{1-\alpha}{\alpha}\mathbf{B}_k},$$

which in turn yield that

$$\frac{\mathbf{B}_k}{\mathbf{B}_j} = \frac{\mathbf{A}_k}{\mathbf{A}_j}\left( \frac{1-\alpha}{\alpha}\mathbf{B}_k + 1 \right) \left( \frac{\frac{\mathbf{A}_k}{\mathbf{A}_j}\left( \frac{1-\alpha}{\alpha}\mathbf{B}_k + 1 \right) - \frac{1-\alpha}{\alpha}\mathbf{B}_k}{\frac{\mathbf{A}_k}{\mathbf{A}_j}\left( \frac{1-\alpha}{\alpha}\mathbf{B}_k + 1 \right)} \right)$$

$$= \frac{\mathbf{A}_k}{\mathbf{A}_j}\left( \frac{1-\alpha}{\alpha}\mathbf{B}_k + 1 \right) - \frac{1-\alpha}{\alpha}\mathbf{B}_k.$$

Now, observe that $0 = \alpha - \left( \left( \frac{\mathbf{B}_k}{\mathbf{B}_j} - \frac{\mathbf{A}_k}{\mathbf{A}_j} \right) \frac{\mathbf{A}_j}{\mathbf{B}_k(\mathbf{A}_k - \mathbf{A}_j)} + 1 \right)^{-1}$ yield that

$$\frac{\mathbf{B}_k}{\mathbf{B}_j} = \frac{1 - \alpha}{\alpha} \frac{\mathbf{B}_k(\mathbf{A}_k - \mathbf{A}_j)}{\mathbf{A}_j} + \frac{\mathbf{A}_k}{\mathbf{A}_j} = \frac{\mathbf{A}_k}{\mathbf{A}_j} \left( \frac{1 - \alpha}{\alpha} \mathbf{B}_k + 1 \right) - \frac{1 - \alpha}{\alpha} \mathbf{B}_k.$$

Thus, similar calculations with $>$ and $<$ instead of $=$, completes the claim. $\qquad\square$

**Theorem A.4.** *Fix $\tau \geq 2$, $\lambda > 0$ and $\alpha^{(2)}, \ldots, \alpha^{(\tau-1)} \in \mathbb{R}$, then*

$$
\begin{aligned}
\alpha^{\star(\tau)} &= \underset{\alpha^{(\tau)} \in \mathbb{R}}{\operatorname{argmin}} \left\| \tilde{\mathbf{y}} - f(\tilde{\mathbf{X}}, \hat{\boldsymbol{\beta}}^{(\tau)}) \right\|_2^2 \\
&= \frac{\left( \frac{\partial}{\partial \alpha^{(\tau)}} f(\tilde{\mathbf{X}}, \hat{\boldsymbol{\beta}}^{(\tau)}) \right)^{\mathsf{T}} \left( \tilde{\mathbf{y}} - \tilde{\mathbf{y}}^{(1)} \right)}{\left\| \frac{\partial}{\partial \alpha^{(\tau)}} f(\tilde{\mathbf{X}}, \hat{\boldsymbol{\beta}}^{(\tau)}) \right\|^2} + 1, \\
&= 1 - \frac{\left( \tilde{\mathbf{y}}_{\alpha=0}^{(\tau)} - \tilde{\mathbf{y}}^{(1)} \right)^{\mathsf{T}} \left( \tilde{\mathbf{y}} - \tilde{\mathbf{y}}^{(1)} \right)}{\left\| \tilde{\mathbf{y}}_{\alpha=0}^{(\tau)} - \tilde{\mathbf{y}}^{(1)} \right\|_2^2}
\end{aligned}
$$

*where $\tilde{\mathbf{y}}^{(1)} = f(\tilde{\mathbf{X}}, \hat{\boldsymbol{\beta}}^{(1)})$, and $\tilde{\mathbf{y}}_{\alpha=0}^{(\tau)} = f(\tilde{\mathbf{X}}, \hat{\boldsymbol{\beta}}_{\alpha=0}^{(\tau)})$.*

*Proof.* Let $\mathcal{L}(\alpha^{(\tau)}, \lambda) = \left\| \tilde{\mathbf{y}} - f(\tilde{\mathbf{X}}, \hat{\boldsymbol{\beta}}^{(\tau)}) \right\|^2$, where $f$ depends on $\alpha^{(\tau)}$ and $\lambda$ through $\hat{\boldsymbol{\beta}}^{(\tau)}$. Note that,

$$f(\tilde{\mathbf{X}}, \hat{\boldsymbol{\beta}}^{(\tau)}) = \kappa(\tilde{\mathbf{X}}, \mathbf{X})(\mathbf{K} + \lambda \mathbf{I})^{-1} \left( \alpha^{(\tau)} \mathbf{y} + (1 - \alpha^{(\tau)}) \mathbf{y}^{(\tau-1)} \right)$$

$$\frac{\partial}{\partial \alpha^{(\tau)}} f(\tilde{\mathbf{X}}, \hat{\boldsymbol{\beta}}^{(\tau)}) = \kappa(\tilde{\mathbf{X}}, \mathbf{X})(\mathbf{K} + \lambda \mathbf{I})^{-1} \left( \mathbf{y} - \mathbf{y}^{(\tau-1)} \right).$$

Then for fixed $\lambda > 0$, we have that

$$\frac{\partial}{\partial \alpha^{(\tau)}} \mathcal{L}(\alpha, \lambda) = \left( \frac{\partial}{\partial \alpha^{(\tau)}} f(\tilde{\mathbf{X}}, \hat{\boldsymbol{\beta}}^{(\tau)}) \right)^{\mathsf{T}} \left( 2 f(\tilde{\mathbf{X}}, \hat{\boldsymbol{\beta}}^{(\tau)}) - 2 \tilde{\mathbf{y}} \right)$$

and since we can decompose $f(\tilde{\mathbf{X}}, \hat{\boldsymbol{\beta}}^{(\tau)})$ as

$$
\begin{aligned}
f(\tilde{\mathbf{X}}, \hat{\boldsymbol{\beta}}^{(\tau)}) &= \alpha^{(\tau)} \kappa(\tilde{\mathbf{X}}, \mathbf{X})(\mathbf{K} + \lambda \mathbf{I})^{-1} \left( \mathbf{y} - \mathbf{y}^{(\tau-1)} \right) + \kappa(\tilde{\mathbf{X}}, \mathbf{X})(\mathbf{K} + \lambda \mathbf{I})^{-1} \mathbf{y}^{(\tau-1)} \\
&= \alpha^{(\tau)} \frac{\partial}{\partial \alpha^{(\tau)}} f(\tilde{\mathbf{X}}, \hat{\boldsymbol{\beta}}^{(\tau)}) + \kappa(\tilde{\mathbf{X}}, \mathbf{X})(\mathbf{K} + \lambda \mathbf{I})^{-1} \mathbf{y}^{(\tau-1)},
\end{aligned}
$$

and set $\frac{\partial}{\partial \alpha^{(\tau)}} \mathcal{L}(\alpha^{(\tau)}, \lambda) = 0$, we can solve as follows

$$
\begin{aligned}
\left( \partial f^{(\tau)} \right)^{\mathsf{T}} \tilde{\mathbf{y}} - \left( \partial f^{(\tau)} \right)^{\mathsf{T}} \kappa(\tilde{\mathbf{X}}, \mathbf{X})(\mathbf{K} + \lambda \mathbf{I})^{-1} \mathbf{y}^{(\tau-1)} &= \alpha^{(\tau)} \left( \partial f^{(\tau)} \right)^{\mathsf{T}} \left( \partial f^{(\tau)} \right) \\
&= \alpha^{(\tau)} \left\| \partial f^{(\tau)} \right\|^2,
\end{aligned}
$$

where we use the notation $\partial f^{(\tau)} \stackrel{\text{def}}{=} \frac{\partial}{\partial \alpha^{(\tau)}} f(\tilde{\mathbf{X}}, \hat{\boldsymbol{\beta}}^{(\tau)})$ for brevity. Now since

$$-\kappa(\tilde{\mathbf{X}}, \mathbf{X})(\mathbf{K} + \lambda \mathbf{I})^{-1} \mathbf{y}^{(\tau-1)} = \kappa(\tilde{\mathbf{X}}, \mathbf{X})(\mathbf{K} + \lambda \mathbf{I})^{-1}(\mathbf{y} - \mathbf{y}^{(\tau-1)}) - \kappa(\tilde{\mathbf{X}}, \mathbf{X})(\mathbf{K} + \lambda \mathbf{I})^{-1} \mathbf{y},$$

we can finalize the proof with

$$\alpha^{\star(\tau)} = \frac{\left( \frac{\partial}{\partial \alpha} f(\tilde{\mathbf{X}}, \hat{\boldsymbol{\beta}}^{(\tau)}) \right)^{\mathsf{T}} \left( \tilde{\mathbf{y}} - \tilde{\mathbf{y}}^{(1)} \right)}{\left\| \frac{\partial}{\partial \alpha} f(\tilde{\mathbf{X}}, \hat{\boldsymbol{\beta}}^{(\tau)}) \right\|^2} + 1,$$

and noting that $\frac{\partial}{\partial \alpha^{(\tau)}} f(\tilde{\mathbf{X}}, \hat{\boldsymbol{\beta}}^{(\tau)}) = \tilde{\mathbf{y}}^{(1)} - \tilde{\mathbf{y}}_{\alpha=0}^{(\tau)}$. $\qquad\square$

Note, in the following we state and prove a slightly more general result than Theorem 4.5.

**Theorem A.5.** *Let $\mathbf{y}^{(\tau)}, \hat{\boldsymbol{\beta}}^{(\tau)}$, and $f(\cdot, \hat{\boldsymbol{\beta}}^{(\tau)})$ be defined as above, and $\alpha \in [0, 1]$, then the following limits hold*

$$\mathbf{y}^{(\infty)} \overset{\text{def}}{=} \lim_{\tau \to \infty} \mathbf{y}^{(\tau)} = \alpha \mathbf{K} (\alpha \mathbf{K} + \lambda \mathbf{I}_n)^{-1} \mathbf{y}$$

$$f(\mathbf{x}, \hat{\boldsymbol{\beta}}^{(\infty)}) \overset{\text{def}}{=} \lim_{\tau \to \infty} f(\mathbf{x}, \hat{\boldsymbol{\beta}}^{(\tau)}) = \alpha \kappa(\mathbf{x}, \mathbf{X})^{\mathsf{T}} (\mathbf{K} + \lambda \mathbf{I}_n)^{-1} \left( \mathbf{I}_n + (1 - \alpha) \mathbf{K} (\alpha \mathbf{K} + \lambda \mathbf{I}_n)^{-1} \right) \mathbf{y}$$

*and if $\alpha > 0$, then*

$$\mathbf{y}^{(\infty)} = \mathbf{K} \left( \mathbf{K} + \frac{\lambda}{\alpha} \mathbf{I}_n \right)^{-1} \mathbf{y}$$

$$f(\mathbf{x}, \hat{\boldsymbol{\beta}}^{(\infty)}) = \alpha f(\mathbf{x}, \hat{\boldsymbol{\beta}}^{(1)}) + (1 - \alpha) f(\mathbf{x}, \hat{\boldsymbol{\gamma}}^{(\infty)})$$

*where* (17) *corresponds to* classical *kernel ridge regression with amplified regularization parameter $\frac{\lambda}{\alpha}$, and we let $\hat{\boldsymbol{\gamma}}^{(\infty)}$ denote the kernel ridge regression parameter associated with solving another kernel ridge regression on the targets $\mathbf{y}^{(\infty)}$ with regularization parameter $\lambda$. Furthermore, the convergence $\lim_{\tau \to \infty} \mathbf{y}^{(\tau)}$ is of linear rate.*

*Proof.* By (9) we have that $\mathbf{K}(\mathbf{K} + \lambda \mathbf{I}_n)^{-1} = \mathbf{V} \mathbf{D} (\mathbf{D} + \lambda \mathbf{I}_n)^{-1} \mathbf{V}^{\mathsf{T}}$ where $\lambda > 0$, $\mathbf{D}$ is positive diagonal and $\mathbf{V}$ is orthogonal, and hence, the eigenvalues of $\mathbf{K}(\mathbf{K} + \lambda \mathbf{I}_n)^{-1}$ are all smaller than 1 in absolute value, and thus $(1 - \alpha)^{\tau-1} \left( \mathbf{K}(\mathbf{K} + \lambda \mathbf{I}_n)^{-1} \right)^{\tau}$ converge to the zero-matrix when $\tau \to \infty$. Thus, using the limit for a geometric series of matrices we get that

$$\lim_{\tau \to \infty} \mathbf{y}^{(\tau)} = \left( \frac{\alpha}{1 - \alpha} \sum_{i=1}^{\infty} \left( (1 - \alpha) \mathbf{K} (\mathbf{K} + \lambda \mathbf{I}_n)^{-1} \right)^i \right) \mathbf{y}$$

$$= \frac{\alpha}{1 - \alpha} (1 - \alpha) \mathbf{K} (\mathbf{K} + \lambda \mathbf{I}_n)^{-1} (\mathbf{I}_n - (1 - \alpha) \mathbf{K} (\mathbf{K} + \lambda \mathbf{I}_n)^{-1})^{-1} \mathbf{y}$$

$$= \alpha \mathbf{K} (\alpha \mathbf{K} + \lambda \mathbf{I}_n)^{-1} \mathbf{y}.$$

If $\alpha > 0$, the remaining result for $\lim_{\tau \to \infty} \mathbf{y}^{(\tau)}$ follows directly. Now, by inserting $\mathbf{y}^{(\infty)}$ and manipulating the result, we get that

$$f(\mathbf{x}, \boldsymbol{\beta}^{(\infty)}) = \kappa(\mathbf{x}, \mathbf{X})^{\mathsf{T}} (\mathbf{K} + \lambda \mathbf{I}_n)^{-1} \left( \alpha \mathbf{y} + (1 - \alpha) \mathbf{y}^{(\infty)} \right)$$

$$= \kappa(\mathbf{x}, \mathbf{X})^{\mathsf{T}} (\mathbf{K} + \lambda \mathbf{I}_n)^{-1} \left( \alpha \mathbf{I}_n + (1 - \alpha) \alpha \mathbf{K} (\alpha \mathbf{K} + \lambda \mathbf{I}_n)^{-1} \right) \mathbf{y}$$

$$= \alpha \kappa(\mathbf{x}, \mathbf{X})^{\mathsf{T}} (\mathbf{K} + \lambda \mathbf{I}_n)^{-1} \left( \mathbf{I}_n + (1 - \alpha) \mathbf{K} (\alpha \mathbf{K} + \lambda \mathbf{I}_n)^{-1} \right) \mathbf{y},$$

and if $\alpha > 0$, then

$$f(\mathbf{x}, \boldsymbol{\beta}^{(\infty)}) = \alpha f(\mathbf{x}, \boldsymbol{\beta}^{(1)}) + (1 - \alpha) \kappa(\mathbf{x}, \mathbf{X})^{\mathsf{T}} (\mathbf{K} + \lambda \mathbf{I}_n)^{-1} \mathbf{K} \left( \mathbf{K} + \frac{\lambda}{\alpha} \mathbf{I}_n \right)^{-1} \mathbf{y}$$

$$= \alpha f(\mathbf{x}, \boldsymbol{\beta}^{(1)}) + (1 - \alpha) f(\mathbf{x}, \hat{\boldsymbol{\gamma}}^{(\infty)}),$$

where we let $\hat{\boldsymbol{\gamma}}^{(\infty)}$ denote the kernel ridge regression parameter associated with the classical kernel ridge regression problem on the targets $\mathbf{y}^{(\infty)}$ with regularization parameter $\lambda$.

Finally, denote by $\mathbf{C} \overset{\text{def}}{=} (1 - \alpha) \mathbf{K} (\mathbf{K} + \lambda \mathbf{I}_n)^{-1}$, then we have that

$$\mathbf{E}(t) \overset{\text{def}}{=} \sum_{i=1}^{t} \mathbf{C}^i - \sum_{i=1}^{\infty} \mathbf{C}^i = \mathbf{C}^{t+1} (\mathbf{C} - \mathbf{I}_n)^{-1},$$

and thus for an additional $s$ steps we have $\mathbf{E}(t + s) = \mathbf{C}^{t+s+1} (\mathbf{C} - \mathbf{I}_n)^{-1} = \mathbf{C}^s \mathbf{E}(t)$. Hence, the convergence is of linear rate as claimed. $\square$

# B Experiments

In the following we show empirical results of performing a simple self-distillation procedure with deep neural networks with varying choices of $\alpha$ to investigate the large scale effects. The experiments are adapted from Mobahi et al. (2020) with the additional introduction of the $\alpha$-parameter. For stronger baselines of the possible performance gains from self-distillation see e.g. Furlanello et al. (2018); Tian et al. (2020b); Ahn et al. (2019); Yang et al. (2018). The following sections provide additional details to that of Section 5.

## B.1 Experimental Setup

We perform self-distillation with ResNet-50 (He et al., 2016) networks on CIFAR-10 (Krizhevsky and Hinton, 2009), with minor pre-processing and augmentations.[15] The model is initialized randomly at each step[16] and trained as described in Section 5 with either estimated optimal parameters, $\hat{\alpha}^{(\tau)}$, or fixed $\alpha$ for all steps. We use Adam optimizer with a learning rate of $10^{-4}$, $\ell_2$ regularization with regularization coefficient $10^{-4}$, and train on the full 50000 training images and validate our generalization performance on the 10000 test images. We use the weights from the last step of optimization at each distillation step for the next step, irrespective of whether a better model occurred earlier in the training. Our models are trained for a fixed 75 epochs, which does not allow our models to overfit the training data, which is important for our models to be suitable for distillation procedures (Dong et al., 2019). The experiments are performed on a single Nvidia Tesla V100 16GB GPU with the PyTorch Lightning framework (Falcon, 2019).

## B.2 Results

We repeat our experiment 4 times and illustrate the mean, minimum and maximum at each distillation step in Figure 5. Each experiment is 11 chains of distillation steps, corresponding to $\alpha \in \{0.0, 0.1, \ldots, 0.9\}$, with the first model initialized identically across all chains. The accuracy reported at the $\tau$'th step is based on comparing the training and validation predictions, $\mathbf{Y}^{(\tau)}$ and $f(\tilde{\mathbf{X}}, \hat{\boldsymbol{\beta}}^{(\tau)})$ with the original training and validation targets; $\mathbf{Y}$ and $\tilde{\mathbf{Y}}$.

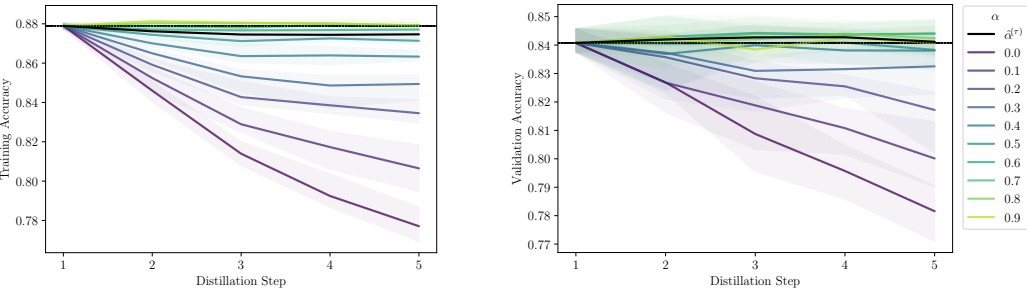

Figure 6: (Identical to Figure 5) Training and validation accuracy for five distillation steps with ResNet-50 models on CIFAR-10. Comparing fixed $\alpha^{(t)}$ for $t = 2 \ldots, \tau$ and estimating optimal weight with $\hat{\alpha}^{(t)}$ at each step. The experiment is repeated four times and the mean (and max/min in shaded) is reported.

The theory introduced in Section 4 suggests that self-distillation corresponds to a progressively amplified regularization of the solution, and larger $\alpha$ dampens the amount of regularization imposed by the procedure more than small values of $\alpha$. Thus, for small $\alpha$ we expect the training accuracy to decrease with each distillation, but for larger $\alpha$ we might experience an increase in training accuracy, due to additional training iterations and a sufficient amount of ground-truth target information being kept in the optimization problem, which could prove beneficial. However, depending on the need for increased regularization, we expect the validation accuracy to increase for some $\alpha$, and possibly decrease for $\alpha$ values either too small or too large. The above properties are observed in Figure 5,

---

[15] Training: We randomly flip an image horizontally with probability $\frac{1}{2}$, followed by a random $32 \times 32$ crop of the $40 \times 40$ zero padded image. Finally we normalize the image to have mean 0 and standard deviation 1. Validation: We normalize the image with the empirical mean and standard deviation from the training data.

[16] Note, we initialize the models equally across all $\alpha$ for one experiment, but alter the seed for initialization between experiments.

where $\alpha \leq 0.4$ generally overregularize the solution, and performance drops with distillation steps. For $\alpha > 0.4$ the performance generally improves with distillation, but for $\alpha$ close to one, the gains reduce, and a suitably balanced $\alpha$ for this experiment would be in $[0.5, 0.7]$. This aligns well with the optimal $\hat{\alpha}^{(\tau)}$ estimated at approx. $0.6$ for each step of self-distillation.

## C  Illustrative Example

In Figure 7 we show the training and validation loss associated with the illustrative example in Section 4.5. Although subtly, we observe a performance improvement in the first few distillation steps across all three choices of $\alpha$. However, for $\alpha = 0$ the solutions progressively underfit the data as $\tau$ increases further.

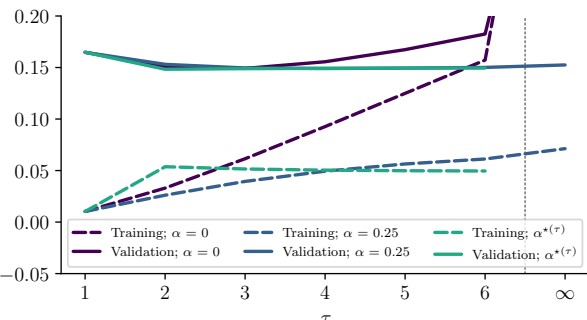

Figure 7: Training (dashed lines) and validation (solid lines) loss associated with $\alpha = 0$, $\alpha = 0.25$, and $\alpha^{\star(\tau)}$. Both losses associated with $\alpha = 0$ for $\tau \to \infty$ are huge compared to finite $\tau$ and the plot is bounded accordingly.

## D  Connection to Neural Networks

This paper theoretically investigates self-distillation of kernel ridge regression models, but distillation procedures are more commonly used in a deep learning setting. However, recent research in the over-parameterized regime has shown great progress and connected wide neural networks with kernel ridge regression using the Neural Tangent Kernel (NTK) (Lee et al., 2019, 2020; Hu et al., 2019). The following is a brief and informal connection between kernel ridge regression and wide[17] neural networks, motivating our problem setup and approach to estimate $\hat{\alpha}^{(\tau)}$ in Section 5.

Consider a neural network with scalar output $f_{\mathrm{nn}}(\mathbf{x}, \boldsymbol{\theta}) \in \mathbb{R}$, where $\boldsymbol{\theta}(t) \in \mathbb{R}^D$ is the vector of all network parameters at training iteration $t \geq 0$, and $\mathbf{x} \in \mathbb{R}^d$ some input. We consider the case where we use gradient descent on the MSE objective, $\mathcal{L}(\boldsymbol{\theta}) = \frac{1}{2} \sum_{i=1}^n (f_{\mathrm{nn}}(\mathbf{x}_i, \boldsymbol{\theta}) - y_i)^2$ over some training dataset $\mathcal{D}_{\mathrm{train}} \subseteq \mathbb{R}^d \times \mathbb{R}$. Consider the first-order Taylor-expansion of $f_{\mathrm{nn}}(\mathbf{x}, \boldsymbol{\theta})$ w.r.t its parameters at initialization, $\boldsymbol{\theta}(0)$,

$$f_{\mathrm{nn}}(\mathbf{x}, \boldsymbol{\theta}) \approx f_{\mathrm{nn}}(\mathbf{x}, \boldsymbol{\theta}(0)) + \langle \nabla_{\boldsymbol{\theta}} f_{\mathrm{nn}}(\mathbf{x}, \boldsymbol{\theta}(0)), \boldsymbol{\theta} - \boldsymbol{\theta}(0) \rangle \tag{19}$$

where $f_{\mathrm{nn}}(\mathbf{x}, \boldsymbol{\theta}(0))$ and $\nabla_{\boldsymbol{\theta}} f_{\mathrm{nn}}(\mathbf{x}, \boldsymbol{\theta}(0))$ are constants w.r.t. $\boldsymbol{\theta}$. For sufficiently wide networks, (19) holds, and we say that we are in the *(NTK) regime* (Arora et al., 2019; Lee et al., 2019). Now, let $\varphi(\mathbf{x}) \stackrel{\text{def}}{=} \nabla_{\boldsymbol{\theta}} f_{\mathrm{nn}}(\mathbf{x}, \boldsymbol{\theta}(0))$ for any $\mathbf{x} \in \mathbb{R}^d$, and denote the random kernel $\kappa(\mathbf{x}_i, \mathbf{x}_j) \stackrel{\text{def}}{=} \langle \varphi(\mathbf{x}_i), \varphi(\mathbf{x}_j) \rangle$ for any $\mathbf{x}_i, \mathbf{x}_j \in \mathbb{R}^d$ (Jacot et al., 2018). For sufficiently wide networks, the random kernel converges to a deterministic kernel, and since the r.h.s. of (19) is linear, one can show that minimizing $\mathcal{L}$ with gradient descent leads to the solution of the kernel regression problem, with the NTK; $\mathbf{x} \mapsto \kappa(\mathbf{x}, \mathbf{X})^\intercal \kappa(\mathbf{X}, \mathbf{X})^{-1} \mathbf{y}$, where $\mathbf{X} \in \mathbb{R}^{n \times d}$ is the matrix of training inputs, and $\mathbf{y} \in \mathbb{R}^n$ the vector of training targets (Arora et al., 2019; Lee et al., 2019). It has been shown that when minimizing the $\ell_2$-regularized MSE loss, the solution becomes the kernel ridge regression solution (Lee et al., 2020).

---

[17]Note, we refer to width as the number of hidden nodes in a fully connected neural network or channels in a convolutional neural network.

The connections between neural networks and kernel ridge regressions in knowledge distillation settings have, to the best of our knowledge, not been explicitly investigated yet, but we hope that the results of this paper will improve the understanding of self-distillation of neural networks once such a connection is made rigorously.

## E    Connections to Constrained Optimization Problem

The setup investigated in this paper is the unconstrained optimization problem presented in (2), but some of the results can easily be extended to a constrained optimization problem, with a general regularization functional in Hilbert spaces, namely the natural extension of the setup proposed by Mobahi et al. (2020). For the rest of this section we assume $\alpha^{(2)} = \cdots = \alpha^{(\tau)} = \alpha$. Mobahi et al. (2020) propose to solve the problem

$$f^{(\tau)} \stackrel{\text{def}}{=} \underset{f \in \mathcal{F}}{\operatorname{argmin}} \int_{\mathcal{X}} \int_{\mathcal{X}} u(\boldsymbol{x}, \boldsymbol{x}') f(\boldsymbol{x}) f(\boldsymbol{x}') d\boldsymbol{x} d\boldsymbol{x}' \quad \text{s.t.} \quad \frac{1}{N} \sum_{n=1}^{N} (f(\boldsymbol{x}_n) - y_n)^2 \leq \varepsilon, \qquad (20)$$

where $\varepsilon > 0$ is a desired loss tolerance, $\tau \geq 1$, $f^{(0)}(\boldsymbol{x}_n) = y_n$ for $n = 1, \ldots, N$, and $u$ being symmetric and such that $\forall f \in \mathcal{F}^{18}$ the double integral is greater than or equal to 0 with equality only when $f(x) = 0$. See Mobahi et al. (2020) for details.

The natural extension of this problem is to include ground-truth labels, and solve the weighted problem

$$f^{(\tau)} = \underset{f \in \mathcal{F}}{\operatorname{argmin}} \int_{\mathcal{X}} \int_{\mathcal{X}} u(\boldsymbol{x}, \boldsymbol{x}') f(\boldsymbol{x}) f(\boldsymbol{x}') d\boldsymbol{x} d\boldsymbol{x}' \qquad (21)$$

$$\text{s.t.} \quad \frac{\alpha}{N} \sum_{n=1}^{N} (f(\boldsymbol{x}_n) - y_n)^2 + \frac{1-\alpha}{N} \sum_{n=1}^{N} \left(f(\boldsymbol{x}_n) - f^{(\tau-1)}(\boldsymbol{x}_n)\right)^2 \leq \varepsilon, \qquad (22)$$

for $\tau \geq 1$, where $\alpha \in [0,1]$ and $f^{(0)}(\boldsymbol{x}_n) = y_n$ for $n = 1, \ldots, N$. In Mobahi et al. (2020), $\alpha = 0$, and this problem completely ignores the ground truth data after the first model fit, and it is easy to see that consecutive self-fits will be penalized increasingly stronger, and eventually collapse to zero, whenever $\frac{1}{N} \sum_{n=1}^{N} (f(\boldsymbol{x}_n) - y_n)^2 \leq \varepsilon$. The case $\alpha = 1$, corresponds to fitting to the ground-truth at each iteration, and do not benefit from distillation, and thus is without interest here.

### E.1    Collapsing and converging conditions

The regularization functional of (21), is clearly minimized by $f_t(\boldsymbol{x}) = 0$, but in order for this to be a solution for some $\tau \geq 1$, it must hold that

$$\frac{\alpha}{N} \|\boldsymbol{y}\|_2^2 + \frac{1-\alpha}{N} \left\|\boldsymbol{y}^{(\tau-1)}\right\|_2^2 \leq \varepsilon,$$

where we use the notation that $\boldsymbol{y}^{(\tau)} = (f^{(\tau)}(\boldsymbol{x}_1), \ldots, f^{(\tau)}(\boldsymbol{x}_N))^T$ and $\boldsymbol{y} = (y_1, \ldots, y_N)$. For $\tau = 1$, this amounts to $\frac{1}{N} \|\boldsymbol{y}\|_2^2 \leq \varepsilon$, and for $\tau > 1$

$$\frac{1-\alpha}{N} \left\|\boldsymbol{y}^{(\tau-1)}\right\|_2^2 \leq \varepsilon - \frac{\alpha}{N} \|\boldsymbol{y}\|_2^2. \qquad (23)$$

But since the l.h.s. is non-negative, it is required that $\frac{\alpha}{N} \|\boldsymbol{y}\|_2^2 \leq \varepsilon$ in order for $f^{(\tau)}(\boldsymbol{x}) = 0$ to be a solution. Hence, we can construct the following settings, that determine the behavior of the solutions:

1. $\frac{1}{N} \|\boldsymbol{y}\|_2^2 \in [0, \varepsilon] \implies \left\|\boldsymbol{y}^{(\tau)}\right\|_2 = 0 \; \forall \tau \geq 1,$         (Collapsed solution)

2. $\frac{1}{N} \|\boldsymbol{y}\|_2^2 \in \left(\varepsilon, \frac{\varepsilon}{\alpha}\right] \implies \exists \underline{\tau} \geq 1$ such that $\begin{cases} \left\|\boldsymbol{y}^{(\tau)}\right\|_2 > 0 & \forall \tau < \underline{\tau}, \\ \left\|\boldsymbol{y}^{(\tau)}\right\|_2 = 0 & \forall \tau \geq \underline{\tau}, \end{cases}$    (Converging to collapsed solution)

3. $\frac{1}{N} \|\boldsymbol{y}\|_2^2 \in \left(\frac{\varepsilon}{\alpha}, \infty\right) \implies \left\|\boldsymbol{y}^{(\tau)}\right\|_2 > 0 \; \forall \tau \geq 1.$    (Converging to non-collapsed solution)

---

[18]For a given $u$ the function space $\mathcal{F}$ is the space of functions $f$ for which the double integral in (20) is bounded.

If we let $\alpha \to 0$ the interval $\left(\varepsilon, \frac{\varepsilon}{\alpha}\right]$ effectively becomes $(\varepsilon, \infty)$, and any solution will collapse at some point (Mobahi et al., 2020). Analogously, if we let $\alpha \to 1$, the interval $\left(\varepsilon, \frac{\varepsilon}{\alpha}\right]$ effectively becomes empty, and all non-collapsed solutions will converge to a non-zero solution. Hence, if $\alpha > 0$, one can obtain non-collapsing convergence with infinite iterations. Furthermore, if we let $\varepsilon \to 0$, then $[0, \varepsilon]$ and $\left(\varepsilon, \frac{\varepsilon}{\alpha}\right]$ will practically collapse to empty intervals, and we will always obtain convergence to non-collapsing solutions, which will correspond to an interpolating solution. For the remainder we assume $\alpha \in (0, 1)$, since the boundary cases are covered in Mobahi et al. (2020) ($\alpha = 0$) or is without interest ($\alpha = 1$). Furthermore, we assume that $\|\mathbf{y}\|_2 > \sqrt{N}\varepsilon$ to avoid a collapsed solution from the beginning. Utilizing the Karush-Kuhn-Tucker (KKT) conditions for this problem, we can rephrase our optimization problem as

$$f^{(\tau)} = \underset{f \in \mathcal{F}}{\operatorname{argmin}} \frac{\alpha}{N} \sum_{n=1}^{N} (f(\boldsymbol{x}_n) - y_n)^2 + \frac{1 - \alpha}{N} \sum_{n=1}^{N} \left(f(\boldsymbol{x}_n) - f^{(\tau-1)}(\boldsymbol{x}_n)\right)^2$$
$$+ \lambda_\tau \int_{\mathcal{X}} \int_{\mathcal{X}} u(\boldsymbol{x}, \boldsymbol{x}') f(\boldsymbol{x}) f(\boldsymbol{x}') d\boldsymbol{x} d\boldsymbol{x}', \tag{24}$$

where $\lambda_\tau \geq 0$. For suitably chosen $\lambda_\tau$, one can show that $f^{(\tau)}$ is an optimal solution to our problem[19].

### E.2 Extending our results

By direct calculations similar to those of Mobahi et al. (2020) one can obtain the closed form solution of (24), but first we will repeat some definitions from Mobahi et al. (2020). Let the Green's Function $g(x, t)$ be such that $\int_{\mathcal{X}} u(x, x') g(x', t) dx' = \delta(x - t)$, where $\delta$ is the Dirac delta, and let $[\mathbf{G}]_{j,k} = \frac{1}{N} g(\mathbf{x}_j, \mathbf{x}_k)$ and $[\mathbf{g}(\mathbf{x})]_k = \frac{1}{N} g(\mathbf{x}, \mathbf{x}_k)$, where $\mathbf{G}$ is a matrix and $\mathbf{g}(\mathbf{x})$ a vector dependent on $\mathbf{x}$. Now we can present the proposition.

**Proposition E.1.** *For any $\tau \geq 1$, the problem* (24) *has a solution of the form*

$$\mathbf{y}^{(\tau)} = \mathbf{g}(\mathbf{x})^\intercal \left(\mathbf{G} + \lambda_\tau \mathbf{I}\right)^{-1} (\alpha \mathbf{y} + (1 - \alpha)\mathbf{y}^{(\tau-1)}),$$

*where* $\mathbf{y}^{(0)} \stackrel{\text{def}}{=} \mathbf{y}$.

Since $\mathbf{G}$ is positive semi-definite, we can decompose it as $\mathbf{G} = \mathbf{V}\mathbf{D}\mathbf{V}^\intercal$. Define $\mathbf{B}^{(0)} \stackrel{\text{def}}{=} \mathbf{I}$, then for $\tau \geq 1$, we have that $\mathbf{y}^{(\tau)} = \mathbf{V}\mathbf{B}^{(\tau)}\mathbf{V}^\intercal \mathbf{y}$, where we set

$$\mathbf{B}^{(\tau)} \stackrel{\text{def}}{=} \frac{\alpha}{1 - \alpha} \sum_{i=1}^{\tau-1} (1 - \alpha)^{\tau-i} \prod_{j=i}^{\tau-1} \mathbf{A}^{(j+1)} + (1 - \alpha)^{\tau-1} \prod_{j=1}^{\tau} \mathbf{A}^{(j)},$$

$$\mathbf{A}^{(\tau)} \stackrel{\text{def}}{=} \mathbf{D}(\mathbf{D} + \lambda_\tau \mathbf{I})^{-1}.$$

By equivalent calculations as those of Lemma 4.2, we have that

$$\mathbf{B}^{(\tau)} = \mathbf{A}^{(\tau)}((1 - \alpha)\mathbf{B}^{(\tau-1)} + \alpha \mathbf{I}_N) \quad \forall \tau \geq 1,$$

and we can now formulate the following theorem, similar to Theorem 4.3.

**Theorem E.2.** *For any pair of diagonals of $\mathbf{D}$, i.e. $d_k$ and $d_j$, where $d_k > d_j$, we have that for all $\tau \geq 1$ and for $\alpha \in (0, 1)$, then*

$$\operatorname{sign}\left(\frac{[\mathbf{B}^{(\tau)}]_{k,k}}{[\mathbf{B}^{(\tau)}]_{j,j}} - \frac{[\mathbf{B}^{(\tau-1)}]_{k,k}}{[\mathbf{B}^{(\tau-1)}]_{j,j}}\right) \tag{25}$$

$$= \operatorname{sign}\left(\left(\left(\frac{[\mathbf{B}^{(\tau-1)}]_{k,k}}{[\mathbf{B}^{(\tau-1)}]_{j,j}} - \frac{[\mathbf{A}^{(\tau)}]_{k,k}}{[\mathbf{A}^{(\tau)}]_{j,j}}\right) \frac{[\mathbf{A}^{(\tau)}]_{j,j}}{[\mathbf{B}^{(\tau-1)}]_{k,k}([\mathbf{A}^{(\tau)}]_{k,k} - [\mathbf{A}^{(\tau)}]_{j,j})} + 1\right)^{-1} - \alpha\right). \tag{26}$$

*Proof.* The proof follows by analogous calculations to the proof of Theorem 4.3. $\qquad \square$

---

[19]See Mobahi et al. (2020) for a detailed argument.

For the case $\alpha = 1$, the results is identical to Theorem 4.3, since no distillation is actually performed. However, the case $\alpha = 0$ is more involved and we refer the reader to Mobahi et al. (2020) for the treatment of this case, since our setup is identical to that of Mobahi et al. (2020) when $\alpha = 0$.

Finally, due to the dependency of $\lambda_\tau$ on the solution from the previous step of distillation in this constrained optimization problem, we are unable to obtain a simple recurrent expression as (6) and a limiting solution as in Theorem 4.5. However, by the results in Section E.1, we can determine, from the norm of the targets, whether the solution collapses or not.