# OpenReview forum: "Even your Teacher Needs Guidance: Ground-Truth Targets Dampen Regularization Imposed by Self-Distillation"
_NeurIPS.cc/2021/Conference — NeurIPS 2021 Poster_

### Official Review · Reviewer_qZC6 · 2021-07-14

**Rating:** 6
**Confidence:** 3

**Summary:**

This paper studies the theory of self-distillation in the kernel ridge regression setting. The authors extend the theoretical results in Mobahi et al. (2020) to further incorporate the ground-truth labels in the distillation objective and highlight that the ground-truth labels serve to dampen the sparsification and regularization of the self-distilled solutions.

**Limitations And Societal Impact:**

The authors discussed the limitation of their work but did not discuss the negative societal impact. To discuss the negative societal impact, the authors can consider some cases where self-distillation can be applied to achieve some malicious goals.

**Main Review:**

This paper provides closed-form solutions of self-distillation in the kernel ridge regression setting. Overall, the entire paper is mostly well-written and easy to follow. It is a natural extension of the theoretical results in Mobahi et al. (2020), though the theoretical novelty is limited given the prior works (the theorems are not hard to derive based on the results of Mobahi et al. (2020)). See detailed comments below.


1. After reading the paper, I am not convinced that incorporating the ground-truth label is essential during the self-distillation step to overcome underfitting. As you show in Theorem 4.5, the weighting parameter $\alpha$ turns out to amplify the regularization parameter $\lambda$. In this sense, we can also smartly tune $\lambda$ on the validation set to achieve the same effect as tuning $\alpha$. Therefore, I don't think that incorporating the ground-truth labels of training data are necessary.


2. The way to estimate the value of $\alpha$ in section 4.3 is not globally optimal, as it does not account for the future steps of self-distillation. I would instead call it a greedy estimation of $\alpha$.


3. For  ResNet-50 on CIFAR-10, why are the best training accuracy only 88% and the best test accuracy only 94%? In my experience, the training accuracy can always reach very close to 100%, and the test accuracy is around ~94%. Please check out this GitHub: https://github.com/kuangliu/pytorch-cifar.


4. For Line 239-240, you mentioned that "As illustrated in Figure 2a for case (a), the regularization imposed by self-distillation initially improves the quality of the solution". To me, I don't see such a trend in Figure 2(a). The curves corresponding to $f_1, ..., f_{\infty}$ become more underfitting as the index value increases. To better illustrate your argument, I would suggest the authors provide a figure of "$\tau$ vs. training loss" (similar to Figure 5).


5. It would be better to state the contribution in the introduction rather than in related works (Line 84-93). Also, the introductions to the notation (Line 96-101) are well-suited to be at the beginning of Section 3 Problem Setup.


6. One minor correction to footnote 4, for some cases (e.g., using Adam optimizer), the equivalence between L2 regularization and weight decay does not hold. See Loshchilov et al. (2017) and Zhang et al. (2018).


Overall, I think this paper is well-written and studies a critical problem in machine learning, considering that there are still relatively few works exploring the theories of knowledge distillation.  However, the theoretical novelty is very limited as most of the theorems are very natural extensions from the results in Mobahi et al. (2020). Also, I am not very convinced that incorporating the ground-truth label is necessary for self-distillation. Therefore, I vote for a weak rejection at the current stage. I am happy to increase my ratings if the authors can address my concerns properly.

References:

1. Loshchilov, Ilya, and Frank Hutter. "Decoupled weight decay regularization." arXiv preprint arXiv:1711.05101 (2017).

2. Zhang, Guodong, et al. "Three mechanisms of weight decay regularization." ICLR 2018.


====Updates=====

The authors' response resolved my concern. I will increase my score from 5 to 6.

**Time Spent Reviewing:**

7

---

> ### Author Response · Authors · 2021-08-06
> **Response to Reviewer qZC6**
>
> We thank you for your rigorous reading and detailed feedback. We are happy that you found the paper well-written, and that it studies a critical problem. We appreciate the detailed comments and will adapt our paper for a final version - please see details below.
>
> 1. *"I am not convinced that incorporating the ground-truth label is essential during the self-distillation step to overcome underfitting."* and *"We can also smartly tune $\lambda$ on the validation set to achieve the same effect as tuning $\alpha$."*
> Concerning the latter point, it is true that in the limit, our self-distillation procedure is equivalent to larger regularization, but for finite steps, the dynamics differ. Previous results have shown that often self-distillation does in fact improve performance after a few steps, and with varying effect of using the ground-truth. We show that although (in the limit) one could just set the regularization parameter higher, the behaviour of the solutions are different for finite steps. The different behaviour comes from the fact that increasing λ would scale all the eigenvalues of the kernel by the same factor, but not change/sparsify the eigenvalues dynamically as self-distillation does. Hence, the interesting elements are in the steps between the infinite solutions and the initial, where the regularization parameter can not be scaled to obtain the same effect.
> Finally, for the limiting case, tuning $\lambda$ is often performed with a grid search, and here we propose an alternative greedy approach using $\alpha$, that can be carried out without a grid search as argued in Section 4.3 and 4.5.
> Concerning the need for ground-truth labels, we do not claim that incorporating the ground-truth label is necessary for self-distillation, but rather observe that it is often done in practice, and we provide some insights into the effect of doing so.
> 2. The estimation is indeed greedy across multiple $\tau$ (globally), but optimal for a single $\tau$. We will change the naming and add a comment in a final version to emphasize this.
> 3. The empirical experiments in our paper are constrained by the theoretical set-up and it appears reasonable to assume that this is the cause of the discrepancy in performance. Indeed, the empirical experiments on CIFAR-10 are done in a highly simple setting; e.g. only minor pre-processing and augmentations, as well as no learning rate schedule. Most importantly we minimize the MSE loss in (18), and not the more common cross-entropy loss. These differences combined likely causes the performance drop compared to experiments with cross-entropy loss, augmentations and learning rate schedules. We will add a comment about the differences in performance and setup in a final version, and if time and compute allows, we will repeat the empirical experiments with both the cross-entropy loss and a learning rate schedule (similarly to the provided link). Although we expect a performance improvement with these changes, we do not expect the overall finding of the effect of $\alpha^{\star(\tau)}$ to be significantly different.
> 4. Although subtly, the performance does in fact improve for the first few distillation steps. We will include a plot of $\tau$ vs. loss for both the training and validation loss for the three examples: $\alpha = 0$, $\alpha = 0.25$ and $\alpha^{\star(\tau)}$ illustrating this observation.
> 5. We agree on these comments and will rearrange the subsections in a final version.
> 6. Agree - we will emphasize this more clearly in a final version.
>
> Once again, we thank you for the valuable feedback, and we hope that you will consider increasing your score.

---

> > ### Comment · Reviewer_qZC6 · 2021-08-25
> > **Further comments**
> >
> > Thanks for the response. Most of my concerns are addressed.
> >
> > **1. Hence, the interesting elements are in the steps between the infinite solutions and the initial, where the regularization parameter can not be scaled to obtain the same effect.**
> > - If I understand correctly, for $\lambda>0$ and $\alpha=0$, the self-distillation step continues to sparsify the basis. This may not be the case for $\alpha>0$, as sometimes the diagonal values of the corresponding basis will increase. Therefore, there exists some intermediate solutions for self-distillation with ground-truth labels (i.e., $\alpha\geq0$) that cannot be reached by self-distillation without ground-truth labels (i.e., $\alpha=0$). If this is the case, I think the contribution of this paper is solid.
> >
> > A quick response will be helpful.

---

> > > ### Author Response · Authors · 2021-08-26
> > > **Response**
> > >
> > > That is correctly understood. Our results precisely show that the dynamics of distillation with ground-truth labels ($\alpha > 0$) differ from the dynamics of distillation without ground-truth labels ($\alpha = 0$), and that the intermediate solutions will differ when ground-truth labels are included.

---

### Official Review · Reviewer_ZxPW · 2021-07-15

**Rating:** 7
**Confidence:** 2

**Summary:**

This paper presented a theoretical and empirical study of weighting the teacher outputs and the target outputs during self distillation. For an iterative self distillation process, a solution is shown to compute the fit at any step of the iteration given the first model. This was further related to classical kernel ridge regression with regularization.

**Ethical Concerns:**

I have no ethical concerns.

**Limitations And Societal Impact:**

There is adequate mention of the limitations of this work. The authors did not address the societal impact, and I agree with the assessment that it is not necessary for this work.

**Main Review:**

The existing theory behind distillation is thin, and this paper makes a step toward a better grasp of it. I found the claims well supported by experiments and well explained in the text. I have not thoroughly checked the proofs in the supplementary material.

I found the writing quality high and the exposition clear.

The findings and theory presented is likely significant in the community's effort to deepen our understanding of distillation.

**Time Spent Reviewing:**

3 hrs.

---

> ### Author Response · Authors · 2021-08-06
> **Response to Reviewer ZxPW**
>
> We thank you for the feedback, and appreciate that you found the writing quality high, the exposition clear, and the findings significant for the understanding of distillation.

---

### Official Review · Reviewer_wK1m · 2021-07-15

**Rating:** 6
**Confidence:** 2

**Summary:**

This paper proposed a variant of self distillation which incorporates both the model output and the ground truth targets. The author provided a detailed theoretical analysis about how fixed weighted ground truth restrain the regularization amplified by self distillation.

**Limitations And Societal Impact:**

There is no negative societal impact of this work.

**Main Review:**

The paper clearly stated the motivation of introducing a weighted ground truth target while self distillation. The theoretical results (Theorem 4.3, 4.5) supported the argument.
The author also provided a close form solution for setting the step wise weight other than a fixed weight for all steps.
The estimation of the optimal weight provided practical usefulness for this method and guidance on how to adjust this hyperparameter efficiently other than grid search.

However, my minor concern about this work are:

the main theoretical results are all derived under the setting of fixed weight for all distillation steps. While in practice the optimal weights are calculated at each step.

In section 4.2, the author argued by introducing the weight parameter, the self distillation process can potentially do not sparsify the basis. According to theorem 4.3, this potential heavily depends on the value of the weights and the weights are derived in a closed form manner at each step. It would be more rigorous if the author could provide some guarantees with probability of theorem 4.3 to show how likely the basis will not sparsify.

Overall, I think this work proposed an inspiring potential method on addressing the evolution of basis problem carried out by Mobahi et al.(2020), along with some practical usage.

**Time Spent Reviewing:**

5

---

> ### Author Response · Authors · 2021-08-06
> **Response to Reviewer wK1m**
>
> We appreciate your careful reading and valuable feedback. We are pleased that you found the motivation clear, and our method of estimating the optimal weighting, in practice, inspiring.
> Regarding your minor concerns:
>
> - "*It would be more rigorous if the author could provide some guarantees with probability of theorem 4.3 to show how likely the basis will not sparsify."*
> We agree that a probabilistic guarantee would be a valuable contribution, but due to the complicated dependence on the data, $\lambda$ and $\alpha$, we are currently not aware of how to provide such a result. However, this is an interesting direction for future research!
> - "The main *theoretical results are all derived under the setting of fixed weight for all distillation steps. While in practice the optimal weights are calculated at each step."*
> Firstly, while Lemma 4.2, Thm. 4.3,  and Thm. 4.5 are derived for fixed $\alpha$, Thm. 4.1 and Thm. 4.4 do not require a fixed $\alpha$. Secondly, we expect Thm. 4.5 to be valid as long as $\alpha^{(\tau)}$ form a convergent series, but we have not been able to prove that e.g. $\hat{\alpha}^{\star(\tau)}$ form such a convergent series.
>
> We thank you for highlighting your minor concerns!

---

### Decision · Program_Chairs · 2021-09-27

**Decision:**

Accept (Poster)

**Comment:**

This paper provides an in-depth analysis of self-distillation in kernel regression setting, and studies the effect of using a weighted combination of ground truth labels and predictions made by the model, to define the next set of target values. It provides a closed form solution for the optimal choice of weighting parameter at each step, and shows how to efficiently estimate this weighting parameter for deep learning and significantly reduce the computational requirements compared to a grid search.

All reviewers find this paper interesting, and rate it in the accept zone. Some of the initial scores were lower at first, but after author's reply, reviewers found them convincing and increased the rating. In concordance with the reviewers, I find distillation an important tool with increasing popularity, and providing a solid understanding about this technique is of great interest to the community.